# Beyond Bradley-Terry Models: A General Preference Model for Language Model Alignment

**Yifan Zhang** [* 1] **Ge Zhang** [* 2] **Yue Wu** [* 3] **Kangping Xu** [1] **Quanquan Gu** [3]

## Abstract

Modeling human preferences is crucial for aligning foundation models with human values. Traditional reward modeling methods, such as the Bradley-Terry (BT) reward model, fall short in expressiveness, particularly in addressing intransitive preferences. In this paper, we introduce *preference embedding*, an approach that embeds responses into a latent space to capture intricate preference structures efficiently, achieving linear query complexity. Additionally, we propose preference score-based General Preference Optimization (GPO), which generalizes reward-based reinforcement learning from human feedback (RLHF). Experimental results show that our General Preference embedding Model (GPM) consistently outperforms the BT reward model on the RewardBench benchmark and effectively models cyclic preferences where any BT reward model behaves like a random guess. Furthermore, evaluations on downstream tasks such as AlpacaEval2.0, following the language model post-training with GPO and our general preference model, reveal performance improvements over BT models. These findings indicate that our method may enhance the alignment of foundation models with nuanced human values. The code is available at https://github.com/general-preference/general-preference-model.

## 1. Introduction

Modeling human preferences is a cornerstone in developing foundation models that interact seamlessly with users. In natural language modeling and reinforcement learning, aligning models with human intent and values has led to significant advancements, including improved text generation and enhanced decision-making policies (Ouyang et al., 2022; Christiano et al., 2017). Traditional approaches often rely on reward modeling, wherein a reward function is learned to guide the optimization of policies. While effective in certain contexts, these methods face expressiveness and computational efficiency challenges, particularly when addressing complex or intransitive human preferences (Tversky, 1969; Munos et al., 2023).

Preference learning algorithms typically employ pairwise comparisons to capture human judgments (Ibarz et al., 2018; Ziegler et al., 2019). The Bradley-Terry (BT) model (Bradley & Terry, 1952) is popular for modeling such pairwise preferences due to its simplicity and computational efficiency: given $K$ responses, a BT reward model cost $\mathcal{O}(K)$ inference-time compute to output the reward dictating the preferences. The efficiency of the BT model comes from the implicit assumption that each option can be conveniently represented by a scalar reward, which inevitably limits the model's capacity to capture the richness of human judgments that may be context-dependent or exhibit intransitivity (Gardner, 1970).

On the other hand, supervised (sequential-classification) pair preference models (PairRM / PairPM) (Jiang et al., 2023; Dong et al., 2024) that predict the preference given a concatenation of the two responses can express complex and intransitive (cyclic) structures. But to fully capture the preference relations among $K$ responses, it requires evaluating $\mathcal{O}(K^2)$ pairwise preferences between all $K$ candidate responses (Munos et al., 2023; Wu et al., 2024b). This quadratic scaling hinders them for applications with larger response sets especially in test-time scaling for reasoning tasks using verifiers and ranking models (Snell et al., 2024; Wu et al., 2024a).

In addition to computational inefficiency, supervised preference models exhibit asymmetric preference behaviors related to positions. The model's design choice can also be highly ad hoc, varying among different templates and model architecture designs.

Based on the above observations, it is thus natural to raise

---

[*]Equal contribution [1]IIIS, Tsinghua University, Beijing, China [2]Shanghai Qi Zhi Institute, Shanghai, China [3]Department of Computer Science, University of California, Los Angeles, California, USA. Correspondence to: Quanquan Gu <qgu@cs.ucla.edu>.

*Proceedings of the $42^{nd}$ International Conference on Machine Learning*, Vancouver, Canada. PMLR 267, 2025. Copyright 2025 by the author(s).

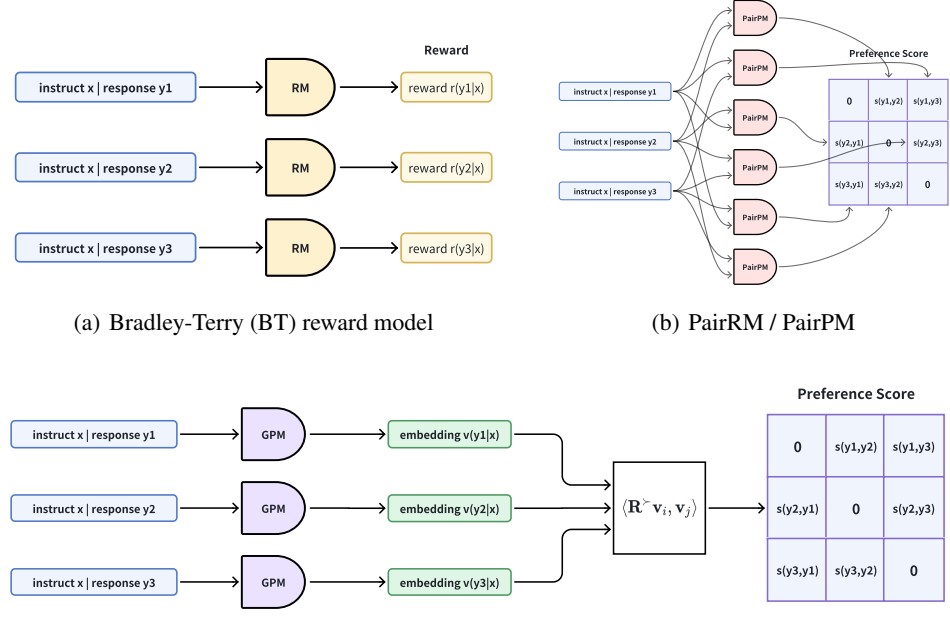

(a) Bradley-Terry (BT) reward model

(b) PairRM / PairPM

(c) General Preference embedding model (GPM)

*Figure 1.* Illustration of (a) Bradley Terry (BT) reward model, (b) supervised pair preference model (PairRM, PairPM) (Jiang et al., 2023; Dong et al., 2024), and (c) our General Preference embedding Model (GPM).

the following question:

*Is there a principled way to model general preference?*

In this paper, we answer this question affirmatively by proposing *preference embedding*, which bridges the gap between expressiveness and efficiency in general preference modeling. Our method embeds responses into a multi-dimensional latent space that captures the complex preference structure beyond transitive relations while allowing for efficient querying of preferences. Notably, our approach achieves a computational complexity of $\mathcal{O}(K)$, matching the efficiency of the BT model but with enhanced expressiveness.

The main contributions of our work are summarized as follows:

- We introduce *preference embedding* for general preference modeling, enabling both efficient and expressive representation of human preferences. Our approach generalizes the Bradley-Terry (BT) reward model by embedding responses into a latent space, capturing complex structures, including intransitive preferences. Notably, our General Preference embedding model (GPM) achieves a query complexity of $\mathcal{O}(K)$ for evaluating preferences among $K$ responses which match the complexity of the Bradley-Terry reward model, an improvement over the $\mathcal{O}(K^2)$ complexity of traditional supervised preference models that rely on pairwise inputs (see Section 4).
- We demonstrate GPM's effectiveness across various tasks,

including CyclicPreference (ours) and the renowned RewardBench (Lambert et al., 2024). Specifically, GPM models intransitive (e.g., cyclic) preferences with near-perfect accuracy, whereas the BT reward model performs like random guessing (see Section 6.2). Additionally, GPM consistently outperforms the BT reward model on RewardBench (see Section 6.1).

- For language model alignment, we propose General Preference Optimization (GPO), which leverages the preference scores provided by GPM. The general preference score can also be integrated as a preference signal into a wide range of RLHF and preference optimization methods (Rafailov et al., 2024; Munos et al., 2023; Wu et al., 2024b). Experimental results on AlpacaEval-2.0 reveal that our approach may improve reward-based language model alignment methods (see Section 6.3).

## 2. Related Work

**Reward-Based Reinforcement Learning from Human Feedback (RLHF).** Typical approaches to modeling human preference for language model alignment usually learn a *reward model* from a preference dataset. The human preference is assumed to follow the Bradley-Terry (BT) model (Bradley & Terry, 1952) or the Thurstone model (Thurstone, 2017). LLM policies then are fine-tuned to maximize these scalar reward signals for better alignment (Christiano et al., 2017; Ziegler et al., 2019; Ouyang et al., 2022). Later, the direct preference optimization (DPO) methods are

proposed by Rafailov et al. (2024) only implicitly to learn a reward model represented by an LLM. The human preference is still assumed to follow the Bradley-Terry model. However, the reliance on scalar rewards imposes a total ordering on preferences, which may not reflect the intransitive or stochastic nature of human judgments (Tversky, 1969; Agranov & Ortoleva, 2017).

**Preference-Based Reinforcement Learning from Human Feedback.** Recently, there emerged a line of works that directly estimate the preference probability without imposing a reward-based preference model or any transitivity assumptions (Lou et al., 2022; Wu et al., 2023; Wang et al., 2023) either for preference-based RL or in the context of RLHF. Efforts have been made to optimize policies directly from pair-wise preference comparisons, thereby mitigating the limitations of scalar reward functions (Munos et al., 2023; Swamy et al., 2024; Rosset et al., 2024; Wu et al., 2024b).

*Figure 2.* Intransitiveness in real-world preferences. Left: Food preferences might cycle (Apple $\succ$ Banana $\succ$ Cherry $\succ$ Apple). Right: Rock-Paper-Scissors is a classic intransitive game.

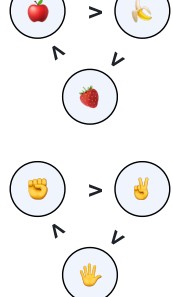

# 3. Background

In this section, we present preliminaries on reward modeling, preference modeling, and reinforcement learning from human feedback (RLHF) for language model alignment. We consider an autoregressive language model that generates responses to the given prompts. Let $\mathbf{x} = [x_1, x_2, \ldots]$ denote a prompt (a sequence of tokens). The language model $\pi$ generates a response $\mathbf{y} = [y_1, y_2, \ldots, y_N]$ based on the conditional probability distribution: $\pi(\mathbf{y} \mid \mathbf{x}) = \prod_{i=1}^{N} \pi(y_i \mid \mathbf{x}, \mathbf{y}_{<i})$, where $\mathbf{y}_{<i}$ represents the sequence of tokens generated before position $i$. In this paper, we assume a general-preference oracle. Given two responses $\mathbf{y}$ and $\mathbf{y}'$ to the same prompt $\mathbf{x}$, the oracle provides the feedback indicating which response is preferred.

$$\mathbb{P}\left(\mathbf{y} \succ \mathbf{y}' \mid \mathbf{x}\right) := \mathbb{E}\left[o\left(\mathbf{y} \succ \mathbf{y}' \mid \mathbf{x}\right)\right].$$

## 3.1. Reward-Based Reinforcement Learning from Human Feedback

The most prevalent approach to aligning language models with human preferences is to consider a scalar reward function $r(\mathbf{y}; \mathbf{x})$ that assigns a numerical score to each response. The preference between two responses is then determined solely by the reward scores for the two responses. For exam-

ple, the Bradley-Terry (BT) model (Bradley & Terry, 1952) is a widely used method for modeling pairwise preferences in this context. However, the BT model can not capture intransitive (e.g. cyclic) preferences effectively (Bertrand et al., 2023). Under BT model, the probability that response $\mathbf{y}$ is preferred over $\mathbf{y}'$ is given by:

$$\mathbb{P}(\mathbf{y} \succ \mathbf{y}' \mid \mathbf{x}) = \sigma\big(r(\mathbf{y}; \mathbf{x}) - r(\mathbf{y}'; \mathbf{x})\big),$$

where $\sigma(z) = 1/(1 + e^{-z})$ is the logistic (sigmoid) function.

In practice, the reward function $r(\mathbf{y}; \mathbf{x})$ is learned by maximizing the likelihood of the observed preference data. Once the reward function is established, policy optimization techniques, such as Proximal Policy Optimization (PPO) (Schulman et al., 2017), can be applied to adjust the language model to generate responses that maximize expected rewards. The optimization problem can be formulated as:

$$\max_{\theta} \mathbb{E}_{\mathbf{x} \sim \mathcal{X}, \, \mathbf{y} \sim \pi_{\theta}(\cdot | \mathbf{x})} \left[r(\mathbf{y}; \mathbf{x})\right] - $$
$$\beta \mathbb{E}_{\mathbf{x} \sim \mathcal{X}} \left[\mathrm{KL}\left(\pi_{\theta}(\cdot \mid \mathbf{x}) \, \| \, \pi_{\mathrm{ref}}(\cdot \mid \mathbf{x})\right)\right],$$

where $\theta$ are the parameters of the policy $\pi_{\theta}$, $\pi_{\mathrm{ref}}$ is a reference policy (often the pre-trained or supervised-fine-tuned language model), $\beta$ is a scaling parameter that controls the strength of regularization, and KL denotes the Kullback-Leibler divergence.

## 3.2. Preference Modeling

We consider the scenario where given a prompt $\mathbf{x}$, a set of responses $\{\mathbf{y}_i\}$ is generated, and human preferences over these responses are represented as pairwise probabilities $\mathbb{P}(\mathbf{y}_i \succ \mathbf{y}_j \mid \mathbf{x}) \in (0, 1)$, indicating the likelihood that response $\mathbf{y}_i$ is preferred over $\mathbf{y}_j$ given the prompt $\mathbf{x}$.

To model these preferences, we define a (pairwise) preference score function:

$$s(\mathbf{y}_i \succ \mathbf{y}_j \mid \mathbf{x}) := \log \frac{\mathbb{P}(\mathbf{y}_i \succ \mathbf{y}_j \mid \mathbf{x})}{1 - \mathbb{P}(\mathbf{y}_i \succ \mathbf{y}_j \mid \mathbf{x})},$$

which represents the log-odds of $\mathbf{y}_i$ being preferred over $\mathbf{y}_j$. This score function allows us to express the preference probability as:

$$\mathbb{P}(\mathbf{y}_i \succ \mathbf{y}_j \mid \mathbf{x}) = \sigma\left(s(\mathbf{y}_i \succ \mathbf{y}_j \mid \mathbf{x})\right), \quad (3.1)$$

where $\sigma(z) = 1/(1 + e^{-z})$ is the logistic function. One can see that the BT model is a special case: $s(\mathbf{y}_i \succ \mathbf{y}_j \mid \mathbf{x}) = r(\mathbf{y}_i; \mathbf{x}) - r(\mathbf{y}_j; \mathbf{x})$.

## 3.3. Pair Preference Models

Existing approaches often involve concatenating the prompt and responses with a template and training an LLM-based sequential classifier in a supervised learning manner. For

example, Jiang et al. (2023) simply concatenate the three segments $(\mathbf{x}, \mathbf{y}_1, \mathbf{y}_2)$ sequentially and form a single input sequence with special tokens as separators:

```
' <source> x  <candidate1> y1
 <candidate2> y2 '
```

Then a sequential classification head on the last token is trained to predict the preference. Another example is Munos et al. (2023), which uses the following template for text summarization:

```
'You are an expert summary rater. Given
a piece of text and two of its possible
summaries, output 1 or 2 to indicate
which summary is better.
Text – ⟨text⟩, Summary 1 – ⟨summary1⟩,
Summary 2 – ⟨summary2⟩.
Preferred Summary –'
```

Then use the last logit for an arbitrarily chosen token as $s(\mathbf{y}_1 \succ \mathbf{y}_2 | \mathbf{x})$ for training.

However, due to the language model's position encoding (Press et al., 2021; Su et al., 2024) and the causal attention (Radford et al., 2018; 2019) mechanism not being symmetric, the candidate's order in the concatenation will affect the final prediction results. It is mitigated by randomly shuffling the two responses in the training dataset but the output is still highly asymmetric. Another limitation is that how to represent the preference score can be highly ad-hoc. The two examples above already use different templates and different model architectures (sequential classification v.s. language modeling).

### 3.4. Preference-based Reinforcement Learning from Human Feedback

To address the potential intransitive human preference, the preference-based LLM alignment algorithms (Munos et al., 2023; Azar et al., 2023; Wu et al., 2024b; Rosset et al., 2024) have been proposed to directly work on the preference pairs instead of assuming a reward function.

Given a preference oracle $\mathbb{P}(\mathbf{y} \succ \mathbf{y}' \mid \mathbf{x})$. The objective is to find a policy $\pi$ that performs well against another competing policy $\pi'$ in terms of these preference probabilities. For example, Azar et al. (2023) consider competing with another fixed policy $\mu$ ($\mathcal{X}$ denotes the distribution over prompts):

$$\max_{\pi} \mathbb{E}_{\mathbf{x}\sim\mathcal{X}} \left[ \mathbb{E}_{\mathbf{y}\sim\pi(\cdot|\mathbf{x}),\, \mathbf{y}'\sim\mu(\cdot|\mathbf{x})} \left[ \mathbb{P}(\mathbf{y} \succ \mathbf{y}' \mid \mathbf{x}) \right] \right.$$
$$\left. - \beta \operatorname{KL}(\pi \| \pi_{\text{ref}}) \right],$$

Other works (Munos et al., 2023; Wu et al., 2024b; Rosset et al., 2024) consider solving the two-player constant-sum game:

$$\max_{\pi} \min_{\pi'} \mathbb{E}_{\mathbf{x}\sim\mathcal{X}} f\left[ \mathbb{E}_{\mathbf{y}\sim\pi(\cdot|\mathbf{x}),\, \mathbf{y}'\sim\pi'(\cdot|\mathbf{x})} \left[ \mathbb{P}(\mathbf{y} \succ \mathbf{y}' \mid \mathbf{x}) \right] \right].$$

To simplify notation, we define the winning probability of a policy $\pi$ over another policy $\pi'$ as:

$$\mathbb{P}(\pi \succ \pi' \mid \mathbf{x}) = \mathbb{E}_{\mathbf{y}\sim\pi(\cdot|\mathbf{x}),\, \mathbf{y}'\sim\pi'(\cdot|\mathbf{x})} \left[ \mathbb{P}(\mathbf{y} \succ \mathbf{y}' \mid \mathbf{x}) \right].$$

The optimization problem then becomes:

$$\max_{\pi} \min_{\pi'} \mathbb{E}_{\mathbf{x}\sim\mathcal{X}} \left[ \mathbb{P}(\pi \succ \pi' \mid \mathbf{x}) \right]. \qquad (3.2)$$

## 4. General Preference Embedding Model

In this section, we propose a general preference embedding framework that can efficiently and expressively model human preferences. Each response is embedded as a vector in a latent space, and the preferences are modeled through interactions between these embeddings using a skew-symmetric operator. We first define preference embeddings, which serve as the foundation for modeling the relationships between responses.

**Definition 4.1** (Preference Embeddings). Given a prompt $\mathbf{x}$, we assign to each response $\mathbf{y}$ a preference embedding vector $\mathbf{v}_{\mathbf{y}|\mathbf{x}} \in \mathbb{R}^{2k}$. These embeddings are designed to capture the features relevant to human preferences beyond what can be represented by scalar rewards.

Next, to model the directional nature of preferences, we introduce the skew-symmetric preference operator, which ensures that the model respects the skew-symmetry (antisymmetry) in preference modeling.

**Definition 4.2** (Skew-symmetric Preference Operator). To capture the directional nature of preferences, we define a skew-symmetric (anti-symmetric) preference operator $\mathbf{R}^{\succ} \in \mathbb{R}^{2k \times 2k}$. Specifically, $\mathbf{R}^{\succ}$ is a block-diagonal matrix consisting of $k$ skew-symmetric blocks of the form (for more discussion, please see Appendix A):

$$\mathbf{R}_l = \begin{bmatrix} 0 & -1 \\ 1 & 0 \end{bmatrix}, \quad l = 1, \ldots, k.$$

An example of $\mathbf{R}^{\succ}$ for $k = 2$ is:

$$\mathbf{R}^{\succ} = \begin{bmatrix} 0 & -1 & 0 & 0 \\ 1 & 0 & 0 & 0 \\ 0 & 0 & 0 & -1 \\ 0 & 0 & 1 & 0 \end{bmatrix}.$$

Finally, we define the preference score, which quantifies the degree to which one response is preferred over another. This score is calculated based on the interaction between the preference embeddings, mediated by the skew-symmetric operator.

**Definition 4.3** (Preference Score). The preference score between two responses $\mathbf{y}_i$ and $\mathbf{y}_j$ using preference embeddings is defined as:

$$s(\mathbf{y}_i \succ \mathbf{y}_j \mid \mathbf{x}) = \langle \mathbf{R}^{\succ} \mathbf{v}_{\mathbf{y}_i \mid \mathbf{x}}, \mathbf{v}_{\mathbf{y}_j \mid \mathbf{x}} \rangle, \qquad (4.1)$$

where $\langle \cdot, \cdot \rangle$ denotes the inner product in $\mathbb{R}^{2k}$. This score captures the anti-symmetric relationship between responses induced by human preferences.

We model the preference probability using the logistic function as defined in (3.1)). Our general preference embedding model (GPM) exhibits two desirable properties:

1. **Skew-symmetry.** The preference score function is skew-symmetric, satisfying:

$$s(\mathbf{y}_i \succ \mathbf{y}_j \mid \mathbf{x}) = -s(\mathbf{y}_j \succ \mathbf{y}_i \mid \mathbf{x}).$$

   This reflects the fact that the preference relation is naturally skew-symmetric: if $\mathbf{y}_i$ is preferred over $\mathbf{y}_j$ with probability $p_{i,j}$, then $\mathbf{y}_j$ is preferred over $\mathbf{y}_i$ with probability $1 - p_{i,j}$.
   Specifically,

$$s(\mathbf{y} \succ \mathbf{y} \mid \mathbf{x}) = \langle \mathbf{R}^{\succ} \mathbf{v}_{\mathbf{y} \mid \mathbf{x}}, \mathbf{v}_{\mathbf{y} \mid \mathbf{x}} \rangle = 0.$$

   This means that a response is neither superior nor inferior to itself.

2. **Magnitude preserving.** The skew-symmetric preference operator does not change the representation vector's magnitude, which makes this operation stable for training and inference.

$$\langle \mathbf{R}^{\succ} \mathbf{v}_{\mathbf{y} \mid \mathbf{x}}, \mathbf{R}^{\succ} \mathbf{v}_{\mathbf{y} \mid \mathbf{x}} \rangle = \langle \mathbf{v}_{\mathbf{y} \mid \mathbf{x}}, \mathbf{v}_{\mathbf{y} \mid \mathbf{x}} \rangle.$$

**Relation to Bradley-Terry Model.** If we set $k = 1$, $\mathbf{v}_{\mathbf{y}} = [r(\mathbf{y} \mid \mathbf{x}), c]^{\top}$, where $c$ is a constant and $c \neq 0$ (e.g., $c = 1$), and $\mathbf{R}^{\succ} = \begin{bmatrix} 0 & -1 \\ 1 & 0 \end{bmatrix}$, then the preference score reduces to:

$$s(\mathbf{y}_i \succ \mathbf{y}_j \mid \mathbf{x}) = c\big(r(\mathbf{y}_i \mid \mathbf{x}) - r(\mathbf{y}_j \mid \mathbf{x})\big),$$

and the preference probability becomes:

$$\mathbb{P}(\mathbf{y}_i \succ \mathbf{y}_j \mid \mathbf{x}) = \sigma\big[c\big(r(\mathbf{y}_i \mid \mathbf{x}) - r(\mathbf{y}_j \mid \mathbf{x})\big)\big],$$

which is exactly the Bradley-Terry (BT) model as a disk game (Balduzzi et al., 2019).

## 4.1. Expressiveness of the Model

Our general preference embedding model is fully expressive for any real skew-symmetric preference matrix (see Appendix A.1 for complex representations interpretation). Specifically, we establish the following theorem (similar results have been proved in Balduzzi et al. (2018)):

**Theorem 4.4** (Expressiveness of Preference Embedding Model). Let $\mathbf{P} \in \mathbb{R}^{k \times k}$ be a real skew-symmetric matrix (i.e., $\mathbf{P} = -\mathbf{P}^{\top}$). Then there exist vectors $\{\mathbf{v}_i\}_{i=1}^{k} \subset \mathbb{R}^{2k}$ and a block-diagonal skew-symmetric matrix $\mathbf{R}^{\succ} \in \mathbb{R}^{2k \times 2k}$, with $\mathbf{R}^{\succ}$ consisting of $k$ blocks of the form:

$$\mathbf{R}_l = \begin{bmatrix} 0 & -1 \\ 1 & 0 \end{bmatrix}, \quad l = 1, \ldots, k,$$

such that:

$$P_{ij} = \mathbf{v}_i^{\top} \mathbf{R}^{\succ} \mathbf{v}_j, \quad \forall i, j.$$

Theorem 4.4 suggests that our preference embedding framework can theoretically model arbitrary complex and potentially intransitive (e.g., cyclic) preference structures (see Appendix C for proofs).

## 4.2. Implementing General Preference Embedding Model

When the preference score matrix $\mathbf{P}$ has an even dimension, i.e., $\mathbf{P} \in \mathbb{R}^{2k \times 2k}$, we have a more interesting interpretation based on spectral decomposition.

**Theorem 4.5.** Let $\mathbf{P} \in \mathbb{R}^{2k \times 2k}$ be a real skew-symmetric matrix (i.e., $\mathbf{P} = -\mathbf{P}^{\top}$). Then there exist embeddings $\{\mathbf{v}_i\}_{i=1}^{2k} \subset \mathbb{R}^{2k}$ and a block-diagonal skew-symmetric matrix $\mathbf{R}^{\succ} \in \mathbb{R}^{2k \times 2k}$, such that:

$$P_{ij} = \mathbf{v}_i^{\top} \mathbf{R}^{\succ} \mathbf{v}_j, \quad \forall i, j.$$

Moreover, the representations $\{\mathbf{v}_i\}$ can be constructed from the orthogonal matrix $\mathbf{U}$ in the decomposition of $\mathbf{P}$, scaled by the square roots of the positive eigenvalues of $\mathbf{P}$.

To effectively capture general preferences while maintaining computational efficiency, we implement our preference embedding model by augmenting an existing language model with two additional components: an eigenvalue scale gate and an eigenvector embedding head. The embeddings $\mathbf{v}_{\mathbf{y} \mid \mathbf{x}}$ are derived from the final hidden state of the language model after processing the prompt $\mathbf{x}$ and response $\mathbf{y}$.

**Eigenvalue Scale Gate.** The eigenvalue scale gate $\mathcal{G}_{\lambda}$ computes context-dependent scaling factors $\{\lambda_l(\mathbf{x})\}$, where $\lambda_l(\mathbf{x}) \geq 0$, based solely on the prompt $\mathbf{x}$:

$$\{\lambda_l(\mathbf{x})\} = \mathcal{G}_{\lambda}(\mathbf{x}).$$

This component models how different preference dimensions are weighted in the context of the given prompt, effectively adjusting the importance of various aspects such as helpfulness, instruction-following, and creativity.

**Eigenvector Embedding Head.** The eigenvector embedding head $\mathcal{E}_{\mathbf{v}}$ generates embeddings $\mathbf{v}_{\mathbf{y} \mid \mathbf{x}}$ for each response $\mathbf{y}$ in the context of the prompt $\mathbf{x}$:

$$\mathbf{v}_{\mathbf{y} \mid \mathbf{x}} = \mathcal{E}_{\mathbf{v}}(\mathbf{x}, \mathbf{y}).$$

These embeddings capture the nuanced characteristics of the responses relevant to human preferences.

**Preference Score.** The preference score between two responses is computed as:

$$s(\mathbf{y}_i \succ \mathbf{y}_j \mid \mathbf{x}) = \mathbf{v}_{\mathbf{y}_i|\mathbf{x}}^\top \mathbf{D}(\mathbf{x}) \mathbf{R}^\succ \mathbf{D}(\mathbf{x}) \mathbf{v}_{\mathbf{y}_j|\mathbf{x}}.$$

where $\mathbf{D}(\mathbf{x})$ is a block-diagonal matrix with blocks $\sqrt{\lambda_l(\mathbf{x})}\mathbf{I}_2$, and $\mathbf{R}^\succ$ is the skew-symmetric preference operator. We normalize the embeddings $\mathbf{v}_{\mathbf{y}|\mathbf{x}}$ to have unit length $\|\mathbf{v}_{\mathbf{y}|\mathbf{x}}\|_2 = 1$ to ensure training stability and boundedness of the preference score (see Appendix C for details on boundedness related to Theorem 5.1).

**Automatic Subspace Discovery.** The use of multiple dimensions in the embeddings allows the model to discover different subspaces corresponding to various preference dimensions automatically. Each pair of dimensions can capture distinct aspects of preferences, such as helpfulness, correctness, or stylistic elements. The context-dependent eigenvalues $\lambda_l(\mathbf{x})$ modulate the contributions of these subspaces based on the prompt, enabling the model to adapt to varying user preferences dynamically.

# 5. General Preference Optimization

**Policy Optimization with Preference Score.** Once we have a general preference model that outputs the preference score $s(\mathbf{y}_i \succ \mathbf{y}_j|\mathbf{x})$ at hand, we aim to find a policy $\pi$ that performs well against an opponent policy $\mu$ in terms of expected preference scores. The optimization problem is formulated as:

$$\max_{\boldsymbol{\theta}} \ \mathbb{E}_{\mathbf{x}} \left[ \mathbb{E}_{\mathbf{y} \sim \pi_{\boldsymbol{\theta}}(\cdot|\mathbf{x}), \ \mathbf{y}' \sim \mu(\cdot|\mathbf{x})} \left[ s(\mathbf{y} \succ \mathbf{y}' \mid \mathbf{x}) \right] \right]$$
$$- \beta \mathbb{E}_{\mathbf{x}} \left[ \mathrm{KL} \left( \pi_{\boldsymbol{\theta}}(\cdot \mid \mathbf{x}) \| \pi_{\mathrm{ref}}(\cdot \mid \mathbf{x}) \right) \right], \quad (5.1)$$

where $\pi_{\mathrm{ref}}$ is a reference policy (e.g., the initial language model), $\mu$ is the opponent policy (usually the same as $\pi_{\mathrm{ref}}$), and $\beta > 0$ is a regularization parameter controlling the divergence from the reference policy. We would like to point out that this formulation is different from the many previous works (Wu et al., 2024b; Swamy et al., 2024; Rosset et al., 2024; Munos et al., 2023; Azar et al., 2023) as they consider maximizing the win rate $\mathbb{P}(\mathbf{y} \succ \mathbf{y}'|\mathbf{x})$, while our formulation is to maximize $s(\mathbf{y} \succ \mathbf{y}'|\mathbf{x}) = \log \frac{\mathbb{P}(\mathbf{y} \succ \mathbf{y}'|\mathbf{x})}{\mathbb{P}(\mathbf{y} \prec \mathbf{y}'|\mathbf{x})}$. Note that $\mathbb{P}(\mathbf{y} \succ \mathbf{y}'|\mathbf{x})$ only varies between 0 and 1, while $s(\mathbf{y} \succ \mathbf{y}'|\mathbf{x})$, can be seen as a generalized version of the reward $r(\mathbf{y}; \mathbf{x})$ in RLHF or DPO (see Section 4), can take arbitrary values.

**General Preference Optimization (GPO).** We consider the iterative preference optimization process such as SPPO (Wu et al., 2024b), while we use preference score instead of preference probability in the loss form. SPPO used $K$ responses for each prompt $\mathbf{x}$ and calculated the empirical win rate of

each response $\mathbf{y}_k$. Instead, we calculate $\widehat{s}(\mathbf{y}_i \succ \mu \mid \mathbf{x})$ to estimate the empirical win rate over the distribution $\mu$ as below:

$$\widehat{s}(\mathbf{y}_i \succ \mu \mid \mathbf{x}) = \frac{1}{K} \sum_{k=1}^{K} s(\mathbf{y}_i \succ \mathbf{y}_k \mid \mathbf{x}), \forall i \in [K],$$
$$(5.2)$$

At each iteration $t$, GPO has the following learning objective:

$$\boldsymbol{\theta}_{t+1} = \arg\min_{\boldsymbol{\theta}} \mathbb{E}_{\mathbf{x} \sim \mathcal{X}, \mathbf{y} \sim \pi_{\boldsymbol{\theta}_t}(\cdot|\mathbf{x})} \left[ \left( \log \frac{\pi_{\boldsymbol{\theta}}(\mathbf{y} \mid \mathbf{x})}{\pi_{\boldsymbol{\theta}_t}(\mathbf{y} \mid \mathbf{x})} \right. \right.$$
$$\left. \left. - \frac{1}{\beta} \left( \widehat{s}(\mathbf{y} \succ \pi_{\boldsymbol{\theta}_t} \mid \mathbf{x}) - \log Z_{\pi_{\boldsymbol{\theta}_t}}(\mathbf{x}) \right) \right)^2 \right], \quad (5.3)$$

where we have the normalizing factor $Z_{\pi_{\boldsymbol{\theta}_t}}(\mathbf{x}) := \sum_{\mathbf{y}} \pi_{\boldsymbol{\theta}_t}(\mathbf{y}|\mathbf{x}) \exp\left(\widehat{s}(\mathbf{y} \succ \pi_{\boldsymbol{\theta}_t} \mid \mathbf{x})\right)$. In practice, we directly replace $\log Z_{\pi_{\boldsymbol{\theta}_t}}(\mathbf{x})$ with $0$[1].

Intuitively, if a response $\mathbf{y}$ receives a high average score, GPO will increase its log probability. We report the empirical performance of GPO in Section 6.3. The following theorem establishes the convergence properties of GPO:

**Theorem 5.1.** Consider the optimization problem defined by the GPO loss (5.3) and assume it is realizable. Let $\{\pi_{\boldsymbol{\theta}_t}\}_{t=1}^T$ denote the sequence of policies generated by GPO, and define $\bar{\pi}_T = \frac{1}{T} \sum_{t=1}^T \pi_{\boldsymbol{\theta}_t}$ as the average policy. Given that the preference score $s$ is bounded within $[-\rho, \rho]$, by setting $\beta = \Theta\left(\sqrt{T}\right)$, we have:

$$\max_{\pi} s\left(\pi \succ \bar{\pi}_T\right) - \min_{\pi} s\left(\pi \prec \bar{\pi}_T\right) = O\left(\frac{1}{\sqrt{T}}\right).$$

**Connection to Policy Gradient.** Applying policy gradient on (5.1) gives:

$$\mathbb{E}_{\mathbf{x} \sim \mathcal{X}, \mathbf{y} \sim \pi_{\boldsymbol{\theta}}} \left[ \left( \widehat{s}(\mathbf{y} \succ \pi_{\boldsymbol{\theta}_t}) - \beta \log \frac{\pi_{\boldsymbol{\theta}}(\mathbf{y}|\mathbf{x})}{\pi_{\boldsymbol{\theta}_t}(\mathbf{y}|\mathbf{x})} \right) \right.$$
$$\left. \nabla_{\boldsymbol{\theta}} \log \pi_{\boldsymbol{\theta}}(\mathbf{y}|\mathbf{x}) \right]$$
$$= \mathbb{E}_{\mathbf{x} \sim \mathcal{X}, \mathbf{y} \sim \pi_{\boldsymbol{\theta}}} \left[ -\nabla_{\boldsymbol{\theta}} \left( \widehat{s}(\mathbf{y} \succ \pi_{\boldsymbol{\theta}_t}) - \beta \log \frac{\pi_{\boldsymbol{\theta}}(\mathbf{y}|\mathbf{x})}{\pi_{\boldsymbol{\theta}_t}(\mathbf{y}|\mathbf{x})} \right)^2 \right].$$

So Equation (5.3) can also be seen as a policy gradient method for the optimization problem (5.1).

**Remark 5.2.** Note that the general preference score given by our GPM can also be integrated as a preference (reward) signal for many other off-the-shelf RLHF and preference optimization methods, including (iterative) DPO-based

---

[1]In late stages of the iterative training, $\pi_{\boldsymbol{\theta}_t}$ is close to equilibrium so the preference model can not distinguish between policy $\pi_{\boldsymbol{\theta}}$ and the opponent policy $\pi_{\boldsymbol{\theta}_t}$( meaning $\widehat{s}(\mathbf{y} \succ \pi_{\boldsymbol{\theta}_t} \mid \mathbf{x}) \approx 0$). Therefore, we have $\log Z_{\pi_{\boldsymbol{\theta}_t}}(\mathbf{x}) \approx 0$.

methods (Rafailov et al., 2024), IPO (Azar et al., 2023), NLHF (Munos et al., 2023), SPPO (Wu et al., 2024b) and REBEL (Gao et al., 2024), as well as PPO-based methods (Ouyang et al., 2022) by directly optimizing problem (5.1).

# 6. Experiments

We conducted several experiments to evaluate the effectiveness of the proposed General Preference embedding Model (GPM) in comparison to traditional reward-based models, particularly focusing on its ability to model general preference and improve language model alignment.

*Table 1.* Comparison of Bradley-Terry (BT) reward models and General Preference embedding models (GPM) on cyclic preference datasets. Cyclic No. 1: `Honest ≻ Truthful ≻ Helpful ≻ Honesty`; Cyclic No. 2: `IF ≻ Truthful ≻ Helpful ≻ IF`; Cyclic No. 3: `IF ≻ Honesty ≻ Helpful ≻ IF`; Cyclic No. 4: `IF ≻ Honesty ≻ Truthful ≻ IF`.

| Model | Dataset | Acc. (%) |
|---|---|---|
| Random Guess | | 50.0 |
| BT RM | Cyclic No. 1 | 62.4 |
| GPM | Cyclic No. 1 | **100.0 (+37.6)** |
| BT RM | Cyclic No. 2 | 61.6 |
| GPM | Cyclic No. 2 | **100.0 (+38.4)** |
| BT RM | Cyclic No. 3 | 50.0 |
| GPM | Cyclic No. 3 | **100.0 (+50.0)** |
| BT RM | Cyclic No. 4 | 62.9 |
| GPM | Cyclic No. 4 | **100.0 (+37.1)** |

## 6.1. Experiments on RewardBench

We compare the GPM and BT reward model on the RewardBench benchmark (Lambert et al., 2024), which covers diverse preference modeling tasks, including Chat, Chat-Hard, Safety, and Reasoning.

**Datasets and Experimental Setup.** We train both BT RMs and GPMs using the decontaminated version of Skywork Reward Data Collection (Liu & Zeng, 2024), which contains around 80k pairwise preference data from tasks in various domains. We evaluate both models on RewardBench, using two different base models: **Gemma-2B-it** (Team et al., 2024) (2B parameters) and **Llama-3.1-8B-Instruct** (Dubey et al., 2024) (8B parameters), which are well-suited for instruction-following tasks (please refer to Appendix E.3 for the implementation details).

**Results and Analysis.** The results are presented in Table 2. On RewardBench, using the Gemma-2B-it base model, GPM achieves an average score of 82.29%, which is an improvement of 7.44% over the BT model's average score of 74.85%. Specifically, in the Chat task, GPM improves performance from 67.32% (BT RM) to 79.61%, and in the Chat-Hard task, from 63.37% to 75.66%. For the Llama-

3.1-8B-Instruct base model, GPM achieves an average score of 91.90% (embedding dimension 8), representing a 1.34% improvement over the BT model's average score of 90.56%. In the Chat task, GPM improves from 88.55% (BT RM) to 93.58%, and in the Chat-Hard task, from 85.75% to 88.38%. These results indicate that GPM generally outperforms the BT model across various base models and tasks, particularly in the Chat and Chat-Hard categories which may involve more nuanced or complex preferences. Note that BT RM is a special case of GPM when the embedding dimension $d = 1$ (see Section 4). Further analysis on challenging cases is in Appendix E.1.

**Ablation studies**. We conducted ablation studies to assess the impact of varying the embedding dimension in GPM. As shown in Table 2, the performance of GPM varies with the embedding dimension. For the Llama-3.1-8B-Instruct base model, an embedding dimension of 8 achieves the highest average score of 91.90%, compared to 91.86% with a dimension of 6 and 91.60% with a dimension of 4. In the Chat-Hard task with the same base model, the highest score of 88.38% is achieved with an embedding dimension of 2, compared to 87.50% with dimension 8. In addition, we can find that for the Gemma-2B-it base model, the highest average score of 82.29% is achieved with an embedding dimension of 6, showing an improvement over lower dimensions, such as 80.43% with dimension 4. These results suggest that the optimal embedding dimensions vary across different base models and tasks. For additional ablation studies on GPM architecture design, please refer to Appendix E.1.

## 6.2. Cyclic Preference Modeling

We evaluate the ability of GPM to capture intransitive, cyclic preferences that traditional transitive models (like the BT model) struggle to represent. Specifically, we evaluate GPMs and BT RMs on CyclicPreference datasets, which are constructed based on the Ultrafeedback dataset (Cui et al., 2024) (See Appendix E).

**Training and Evaluation.** We trained GPMs and BT RMs using the Gemma-2B-it language model as the base and evaluated the models based on their ability to predict intransitive preferences. For GPM, the loss function is Equation (A.1). For the Bradley-Terry (BT) model, the loss function is $\mathcal{L} = -\log \sigma(r_w - r_l)$ (Ouyang et al., 2022). Since cyclic preferences are inherently intransitive, we measure accuracy as the percentage of correctly predicted human preferences, where higher scores indicate better handling of non-transitive preferences. As shown in Table 1, the GP representation model achieves near-perfect accuracy across all datasets, significantly outperforming the BT model (we report the test accuracy on the training dataset but with different comparison pairs used in the training dataset). These results validate GPM's ability to capture complex, cyclic

*Table 2.* Comparison between the Bradley-Terry (BT) models and the General Preference embedding models (GPM) with varying embedding head dimensions on RewardBench. The highest scores are in bold. Note that BT RM is a special case of GPM when embedding dimension $d = 1$ (see Section 4).

| Model | Embed Dim. | Chat | Chat-Hard | Safety | Reasoning | Average |
|---|---|---|---|---|---|---|
| **Base Model: Gemma-2B-it** | | | | | | |
| BT RM | 1 | 67.32 | 63.37 | **85.68** | 83.04 | 74.85 |
| GPM | 2 | 77.37 | 73.46 | 85.00 | 85.50 | 80.33 |
| | 4 | 78.77 | 72.59 | 85.54 | 84.82 | 80.43 |
| | 6 | **79.61** | **75.66** | 85.27 | **88.61** | **82.29 (+7.44)** |
| | 8 | 78.49 | 74.34 | 84.19 | 86.95 | 81.00 |
| **Base Model: Llama-3.1-8B-Instruct** | | | | | | |
| BT RM | 1 | 88.55 | 85.75 | 91.49 | **96.47** | 90.56 |
| GPM | 2 | 91.62 | **88.38** | 90.68 | 94.82 | 91.37 |
| | 4 | 93.30 | 86.18 | 91.22 | 95.69 | 91.60 |
| | 6 | 91.90 | 87.50 | **91.62** | 96.40 | 91.86 |
| | 8 | **93.58** | 87.50 | 91.08 | 95.44 | **91.90 (+1.34)** |

*Table 3.* AlpacaEval 2.0 evaluation results. Base model: Llama3-8B-it. Evaluator: GPT-4-turbo. Results grouped by RM/GPM size (2B/8B) and type (BT/GPM), and training iterations (Iter). Bold entries indicate GPM outperforms BT RM under the same SPPO/GPO setting. Metrics follow AlpacaEval 2.0 standard: **LC. WR** = Length-Controlled Win Rate (%), **WR** = Win Rate (%) against the reference model (GPT-4-turbo), **Avg. Len** = Average response length (tokens).

| Size | Type | Iter | SPPO | | | GPO | | |
|---|---|---|---|---|---|---|---|---|
| | | | LC. WR | WR | Avg. Len | LC. WR | WR | Avg. Len |
| | | base | 23.07 | 23.34 | 1959 | 23.07 | 23.34 | 1959 |
| **2B** | **BT RM** | 1 | 31.95 | 31.59 | 1939 | 34.01 | 33.08 | 1929 |
| | | 2 | 36.00 | 36.77 | 2032 | 38.90 | 39.90 | 2049 |
| | | 3 | 40.01 | 42.12 | 2136 | 42.21 | 44.20 | 2151 |
| | **GPM** | 1 | 30.87 | **32.48 (+0.89)** | 2066 | 35.27 | **37.95 (+4.87)** | 2102 |
| | | 2 | 34.54 | **40.76 (+3.99)** | 2301 | 36.77 | **42.96 (+3.06)** | 2343 |
| | | 3 | 36.06 | **45.61 (+3.49)** | 2498 | 37.74 | **48.25 (+4.05)** | 2582 |
| **8B** | **BT RM** | 1 | 32.20 | 27.83 | 1740 | 36.32 | 30.37 | 1702 |
| | | 2 | 39.75 | 36.95 | 1868 | 41.79 | 40.11 | 1933 |
| | | 3 | 42.55 | 40.92 | 1948 | 40.37 | 38.56 | 1969 |
| | **GPM** | 1 | 33.48 | **30.85 (+3.02)** | 1861 | 36.00 | **33.19 (+2.82)** | 1850 |
| | | 2 | 37.93 | **38.38 (+1.43)** | 2029 | 40.81 | **42.80 (+2.69)** | 2115 |
| | | 3 | 39.45 | **41.64 (+0.72)** | 2385 | 38.98 | **41.54 (+2.98)** | 3249 |

preferences, confirming the theoretical advantages of using a preference embedding-based approach over traditional reward models that assume transitivity.

### 6.3. Downstream Performance on Aligning Language Models with Human Preferences

We further investigate the effectiveness of GPM in language model alignment using Self-Play Policy Optimization (SPPO) (Wu et al., 2024b) and our proposed General Preference Optimization (GPO), integrating preference scores provided by our GP representation model (GPM). We evaluated the models on AlpacaEval 2.0 (Dubois et al., 2024), MT-Bench (Zheng et al., 2023), GSM8K, MMLU, etc., several widely used benchmarks for evaluating LLM alignment.

**Results and Analysis.** The evaluation results on the benchmarks are as follows. For AlpacaEval 2.0, we compared

the generated responses of the aligned models with those of GPT-4o-mini and GPT-4-turbo. The results of the three evaluators are presented in Tables 3 and 5. From Table 3, we observe that both SPPO and GPO demonstrate improved win rates with successive iterations, highlighting the iterative nature of these optimization methods, and GPO consistently outperforms SPPO. In addition, the bolded entries indicate that GPM-integrated methods consistently outperform BT RM-based methods under the same settings on Win Rate (WR).

## 7. Conclusion

In this work, we introduce *preference embedding*, a framework for modeling human preferences that can capture complex, intransitive structures. Our General Preference embedding model (GPM) achieves linear complexity while

can model intricate preference relationships. It consistently outperforms traditional models like Bradley-Terry reward models across various benchmarks, including cyclic preference datasets and real-world tasks from RewardBench. Additionally, incorporating preference scores from GPM into policy optimization methods, such as SPPO and the newly introduced General Preference Optimization (GPO), led to performance improvements in downstream tasks that require alignment with intricate human preferences.

## Impact Statement

This paper presents work to align AI systems with human preferences and values. Improving AI alignment could lead to more reliable and helpful AI systems that better serve human needs while respecting human values. The enhanced ability to model complex, intransitive preferences could result in AI systems that better understand and accommodate nuanced human judgments across different contexts. This advancement underscores the potential of AI alignment algorithms in both technological and societal contexts.

## Acknowledgements

We thank Andrew Yao, Yang Yuan, Kaifeng Lyu, Haodong Wen, and Yifan Luo for helpful discussions and feedback. We also thank the anonymous reviewers for their valuable comments and suggestions, which helped us significantly to improve this paper.

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

# Appendix

# A. More on General Preference Embedding

In this section, we present additional discussion on general preference modeling with preference embeddings.

**Proposition A.1.** For any two vectors $\mathbf{v}_i \in \mathbb{R}^{2k}$ and $\mathbf{v}_j \in \mathbb{R}^{2k}$, if $\mathbf{R} \in \mathbb{R}^{2k \times 2k}$ satisfies the following two properties:

1. Skew-symmetry: $\langle \mathbf{R}\mathbf{v}_i, \mathbf{v}_j \rangle = -\langle \mathbf{R}\mathbf{v}_j, \mathbf{v}_i \rangle$.

2. Magnitude preserving: $\langle \mathbf{R}\mathbf{v}_i, \mathbf{R}\mathbf{v}_i \rangle = \langle \mathbf{v}_i, \mathbf{v}_i \rangle$.

Then $\mathbf{R}$ must be in the form $\mathbf{R} = \mathbf{U}\mathbf{J}\mathbf{U}^\top$, where $\mathbf{U} \in \mathbb{R}^{2k \times 2k}$ is an orthonormal matrix (e.g. identity matrix $\mathbf{I}_{2k}$) and $\mathbf{J}$ is a block-diagonal matrix consisting of $k$ skew-symmetric blocks of the form:

$$\mathbf{J}_l = \begin{bmatrix} 0 & -1 \\ 1 & 0 \end{bmatrix}, \quad l = 1, \ldots, k.$$

## A.1. Complex Embeddings Interpretation

Our model can also be interpreted using complex embeddings. By representing the embeddings as complex vectors $\mathbf{v_y} \in \mathbb{C}^k$, we can express the preference score as:

$$s(\mathbf{y}_i \succ \mathbf{y}_j \mid \mathbf{x}) = \mathrm{Im}\left( \langle \mathbf{v}_{\mathbf{y}_i}, \mathbf{v}_{\mathbf{y}_j} \rangle \right),$$

where $\mathrm{Im}(\cdot)$ denotes the imaginary part, and $\langle \cdot, \cdot \rangle$ is the Hermitian inner product. This formulation captures cyclic and intransitive preferences through the angular relationships between complex presentations.

**Theorem A.2** (Expressiveness of Complex Preference Embeddings). Let $\mathbf{P} \in \mathbb{R}^{k \times k}$ be a real skew-symmetric matrix (i.e., $\mathbf{P} = -\mathbf{P}^\top$). Then, there exist complex vectors $\{\mathbf{v}_i\}_{i=1}^k \subset \mathbb{C}^k$ such that:

$$P_{ij} = \mathrm{Im}\left( \langle \mathbf{v}_i, \mathbf{v}_j \rangle \right), \quad \forall i, j.$$

**Example.** For $k = 1$, let $\mathbf{v_y} = e^{i\theta_\mathbf{y}}$, then:

$$s(\mathbf{y}_i \succ \mathbf{y}_j \mid \mathbf{x}) = \sin(\theta_{\mathbf{y}_i} - \theta_{\mathbf{y}_j}).$$

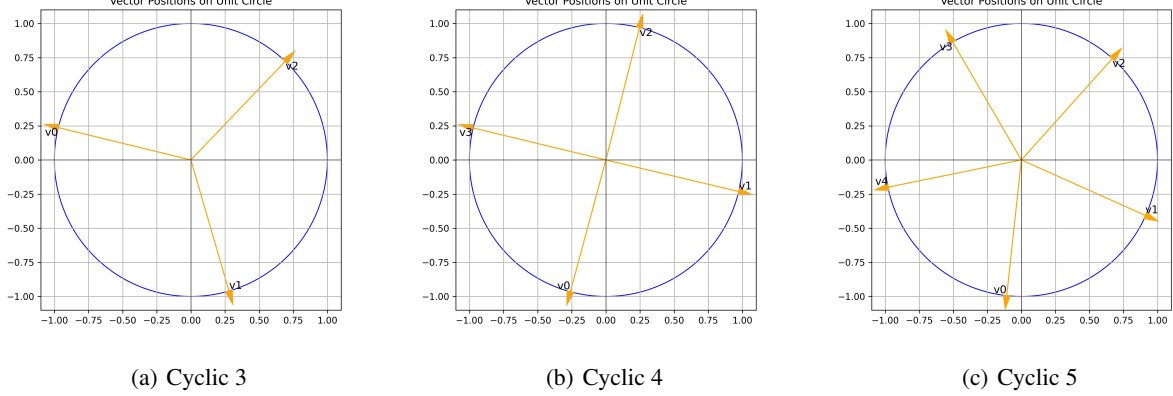

(a) Cyclic 3          (b) Cyclic 4          (c) Cyclic 5

*Figure 3.* Visualization of learned preference embedding vectors for cyclic preferences with sizes 3, 4, and 5, e.g., $A \succ B \succ C \succ A$.

### A.2. Training Objective

The preference embedding can thus be obtained by minimizing the cross-entropy loss over observed preference data. Given a dataset $(\mathbf{x}, \mathbf{y}_w, \mathbf{y}_l) \sim \mathcal{D}$ of preference comparisons, we denote $\mathbb{P}(\mathbf{y}_w \succ \mathbf{y}_l | \mathbf{x})$ as the probability of the winner $\mathbf{y}_w$ being chosen over the loser $\mathbf{y}_l$ (1 if hard preference is given). The cross-entropy loss function is:

$$
\mathcal{L}_{\text{CE}} = - \sum_{(\mathbf{x}, \mathbf{y}_w, \mathbf{y}_l) \in \mathcal{D}} \left[ \mathbb{P}_{\mathcal{D}}(\mathbf{y}_w \succ \mathbf{y}_l \mid \mathbf{x}) \log \sigma \left( \frac{1}{\beta} s(\mathbf{y}_w \succ \mathbf{y}_l \mid \mathbf{x}) \right) \right.
$$
$$
\left. + (1 - \mathbb{P}_{\mathcal{D}}(\mathbf{y}_w \succ \mathbf{y}_l \mid \mathbf{x})) \log \sigma \left( -\frac{1}{\beta} s(\mathbf{y}_w \succ \mathbf{y}_l \mid \mathbf{x}) \right) \right].
$$
(A.1)

Alternatively, if there is an oracle providing continuous scores, we can use a regression loss:

$$
\mathcal{L}_{\text{MSE}} = \sum_{(\mathbf{x}, \mathbf{y}_w, \mathbf{y}_l) \in \mathcal{D}} \left( \frac{1}{\beta} s(\mathbf{y}_w \succ \mathbf{y}_l \mid \mathbf{x}) - s_{\mathcal{D}}(\mathbf{y}_w \succ \mathbf{y}_l \mid \mathbf{x}) \right)^2,
$$

where $s_{\mathcal{D}}(\mathbf{y}_w \succ \mathbf{y}_l \mid \mathbf{x})$ is the dataset-provided score satisfying $\sigma(s_{\mathcal{D}}(\mathbf{y}_w \succ \mathbf{y}_l \mid \mathbf{x})) = \mathbb{P}_{\mathcal{D}}(\mathbf{y}_w \succ \mathbf{y}_l \mid \mathbf{x})$.

## B. More on General Preference Optimization

Note that General Preference Optimization (GPO) employs an iterative framework inspired by the multiplicative weights update (MWU) algorithm (Freund & Schapire, 1999), which update rule is formulated as:

$$
\pi_{t+1}(\mathbf{y} \mid \mathbf{x}) \propto \pi_t(\mathbf{y} \mid \mathbf{x}) \exp(\eta \cdot s(\mathbf{y} \succ \pi_t \mid \mathbf{x})),
$$
$$
t = 1, 2, \ldots,
$$

where $\eta$ denotes the learning rate and $s(\mathbf{y} \succ \pi_t \mid \mathbf{x})$ represents the preference score of response $\mathbf{y}$ over the current policy $\pi_t$ given prompt $\mathbf{x}$.

**The von Neumann winner** represents a fundamental concept in social choice theory (Sen, 1986) that has found significant applications in preference-based reinforcement learning (Owen, 2013; Dudík et al., 2015). It corresponds to the Nash equilibrium of a two-player symmetric game (3.2), representing a mixed strategy—a probability distribution over possible responses—that performs optimally against any opponent in the worst-case scenario.

For notational clarity, we define the preference score of a policy $\pi$ over another policy $\pi'$ as:

$$
s(\pi \succ \pi' \mid \mathbf{x}) = \mathbb{E}_{\mathbf{y} \sim \pi(\cdot | \mathbf{x}), \, \mathbf{y}' \sim \pi'(\cdot | \mathbf{x})} [s(\mathbf{y} \succ \mathbf{y}' \mid \mathbf{x})].
$$

A distribution $\pi^*$ is formally defined as a von Neumann winner when it satisfies:

$$
\min_{\pi' \in \Delta} \mathbb{E}_{\mathbf{x} \sim \mathcal{X}} [s(\pi^* \succ \pi' \mid \mathbf{x})] \geq 0.
$$

This condition ensures that $\pi^*$ is, on average, at least as preferred as any other policy $\pi'$. The symmetric nature of the two-player game (3.2) guarantees the existence of such a winner.

## C. Proofs of Theorems

### C.1. Proof of Proposition A.1.

*Proof.* Let $\mathbf{R} \in \mathbb{R}^{2k \times 2k}$ be a real matrix satisfying the following properties:

1. **Skew-symmetry with respect to the inner product:**

$$
\langle \mathbf{R}\mathbf{v}, \mathbf{w} \rangle = -\langle \mathbf{R}\mathbf{w}, \mathbf{v} \rangle, \quad \forall \mathbf{v}, \mathbf{w} \in \mathbb{R}^{2k}.
$$

2. **Magnitude preserving:**

$$
\langle \mathbf{R}\mathbf{v}, \mathbf{R}\mathbf{v} \rangle = \langle \mathbf{v}, \mathbf{v} \rangle, \quad \forall \mathbf{v} \in \mathbb{R}^{2k}.
$$

Recall that the standard inner product in $\mathbb{R}^{2k}$ is given by $\langle \mathbf{v}, \mathbf{w} \rangle = \mathbf{v}^\top \mathbf{w}$, which is symmetric: $\langle \mathbf{v}, \mathbf{w} \rangle = \langle \mathbf{w}, \mathbf{v} \rangle$.

From the skew-symmetry condition, we have:

$$\langle \mathbf{R}\mathbf{v}, \mathbf{w} \rangle + \langle \mathbf{R}\mathbf{w}, \mathbf{v} \rangle = 0, \quad \forall \mathbf{v}, \mathbf{w} \in \mathbb{R}^{2k}.$$

Since $\langle \mathbf{R}\mathbf{w}, \mathbf{v} \rangle = (\mathbf{R}\mathbf{w})^\top \mathbf{v} = \mathbf{w}^\top \mathbf{R}^\top \mathbf{v}$, the above condition becomes:

$$\mathbf{v}^\top \mathbf{R}^\top \mathbf{w} + \mathbf{w}^\top \mathbf{R}^\top \mathbf{v} = 0, \quad \forall \mathbf{v}, \mathbf{w} \in \mathbb{R}^{2k}.$$

This implies that $\mathbf{R}^\top$ is skew-symmetric:

$$\mathbf{R}^\top = -\mathbf{R}.$$

From the magnitude-preserving property, we have:

$$\langle \mathbf{R}\mathbf{v}, \mathbf{R}\mathbf{v} \rangle = (\mathbf{R}\mathbf{v})^\top \mathbf{R}\mathbf{v} = \mathbf{v}^\top \mathbf{R}^\top \mathbf{R}\mathbf{v} = \mathbf{v}^\top \mathbf{v}, \quad \forall \mathbf{v} \in \mathbb{R}^{2k}.$$

Therefore,

$$\mathbf{R}^\top \mathbf{R} = \mathbf{I}_{2k}.$$

Using $\mathbf{R}^\top = -\mathbf{R}$, we obtain:

$$(-\mathbf{R})\mathbf{R} = \mathbf{I}_{2k} \quad \Rightarrow \quad \mathbf{R}^2 = -\mathbf{I}_{2k}.$$

This shows that $\mathbf{R}$ satisfies the equation $\mathbf{R}^2 = -\mathbf{I}_{2k}$.

The characteristic polynomial of $\mathbf{R}$ is then:

$$\det(\mathbf{R} - \lambda \mathbf{I}_{2k}) = 0.$$

Since $\mathbf{R}^2 = -\mathbf{I}_{2k}$, it follows that the eigenvalues $\lambda$ satisfy:

$$\lambda^2 = -1 \quad \Rightarrow \quad \lambda = \pm i.$$

Thus, $\mathbf{R}$ has eigenvalues $\pm i$, each with algebraic multiplicity $k$.

Because $\mathbf{R}$ is real and skew-symmetric, it can be brought into block-diagonal form via an orthogonal transformation. Specifically, there exists an orthogonal matrix $\mathbf{U} \in \mathbb{R}^{2k \times 2k}$ such that:

$$\mathbf{R} = \mathbf{U}\mathbf{J}\mathbf{U}^\top,$$

where

$$\mathbf{J} = \mathrm{blockdiag}(\mathbf{J}_1, \mathbf{J}_2, \ldots, \mathbf{J}_k),$$

and each block $\mathbf{J}_l$ is a $2 \times 2$ skew-symmetric matrix of the form:

$$\mathbf{J}_l = \begin{bmatrix} 0 & -1 \\ 1 & 0 \end{bmatrix}, \quad l = 1, \ldots, k.$$

This decomposition leverages the standard canonical form for real skew-symmetric matrices, which states that any such matrix can be orthogonally diagonalized into blocks of this type.

Therefore, $\mathbf{R}$ can be expressed as:

$$\mathbf{R} = \mathbf{U}\mathbf{J}\mathbf{U}^\top,$$

where $\mathbf{U} \in \mathbb{R}^{2k \times 2k}$ is an orthogonal matrix, and $\mathbf{J}$ is the block-diagonal matrix consisting of $k$ blocks $\mathbf{J}_l$.

This completes the proof. $\qquad\square$

## C.2. Proof of Theorem 4.4.

*Proof.* We aim to represent the entries of the skew-symmetric matrix $\mathbf{P} \in \mathbb{R}^{k \times k}$ using vectors in $\mathbb{R}^{2k}$ and a block-diagonal skew-symmetric matrix $\mathbf{R}^{\succ} \in \mathbb{R}^{2k \times 2k}$.

For each $i = 1, \ldots, k$, define the vector $\mathbf{v}_i \in \mathbb{R}^{2k}$ as:

$$\mathbf{v}_i = \begin{bmatrix} \mathbf{a}_i \\ \mathbf{b}_i \end{bmatrix},$$

where $\mathbf{a}_i, \mathbf{b}_i \in \mathbb{R}^k$ are real vectors to be specified.

Set $\mathbf{a}_i = \mathbf{e}_i$, the $i$-th standard basis vector in $\mathbb{R}^k$, and define $\mathbf{b}_i$ as:

$$\mathbf{b}_i = \frac{1}{2}\mathbf{p}_i,$$

where $\mathbf{p}_i$ is the $i$-th row of $\mathbf{P}$. Thus, the $j$-th component of $\mathbf{b}_i$ is $(\mathbf{b}_i)_j = \frac{1}{2}P_{ij}$.

Define the block-diagonal matrix $\mathbf{R}^{\succ} \in \mathbb{R}^{2k \times 2k}$ as:

$$\mathbf{R}^{\succ} = \text{blockdiag}(\mathbf{R}_1, \ldots, \mathbf{R}_k),$$

where each block $\mathbf{R}_l$ is the $2 \times 2$ skew-symmetric matrix:

$$\mathbf{R}_l = \begin{bmatrix} 0 & -1 \\ 1 & 0 \end{bmatrix}, \quad l = 1, \ldots, k.$$

Now, compute the inner product $\mathbf{v}_i^\top \mathbf{R}^{\succ} \mathbf{v}_j$:

$$\mathbf{v}_i^\top \mathbf{R}^{\succ} \mathbf{v}_j = \begin{bmatrix} \mathbf{a}_i^\top & \mathbf{b}_i^\top \end{bmatrix} \begin{bmatrix} \mathbf{0}_{k \times k} & -\mathbf{I}_k \\ \mathbf{I}_k & \mathbf{0}_{k \times k} \end{bmatrix} \begin{bmatrix} \mathbf{a}_j \\ \mathbf{b}_j \end{bmatrix} = -\mathbf{a}_i^\top \mathbf{b}_j + \mathbf{b}_i^\top \mathbf{a}_j.$$

Since $\mathbf{a}_i = \mathbf{e}_i$, we have:

$$\mathbf{a}_i^\top \mathbf{b}_j = \mathbf{e}_i^\top \mathbf{b}_j = (\mathbf{b}_j)_i = \frac{1}{2}P_{ji} = -\frac{1}{2}P_{ij}, \tag{C.1}$$

$$\mathbf{b}_i^\top \mathbf{a}_j = \mathbf{b}_i^\top \mathbf{e}_j = (\mathbf{b}_i)_j = \frac{1}{2}P_{ij}. \tag{C.2}$$

Therefore,

$$\mathbf{v}_i^\top \mathbf{R}^{\succ} \mathbf{v}_j = -\left(-\frac{1}{2}P_{ij}\right) + \frac{1}{2}P_{ij} = P_{ij}.$$

Thus, for all $i, j$,

$$P_{ij} = \mathbf{v}_i^\top \mathbf{R}^{\succ} \mathbf{v}_j.$$

This construction shows that any real skew-symmetric matrix $\mathbf{P}$ can be represented in terms of vectors $\{\mathbf{v}_i\} \subset \mathbb{R}^{2k}$ and the block-diagonal skew-symmetric matrix $\mathbf{R}^{\succ}$.

This completes the proof. $\qquad\square$

## C.3. Proof of Theorem 4.5.

*Proof.* Since $\mathbf{P}$ is real and skew-symmetric with even dimension $2k$, it can be brought into block-diagonal form via an orthogonal transformation. Specifically, there exists an orthogonal matrix $\mathbf{U} \in \mathbb{R}^{2k \times 2k}$ such that:

$$\mathbf{P} = \mathbf{U}\mathbf{\Lambda}\mathbf{U}^\top,$$

where $\mathbf{\Lambda}$ is a block-diagonal matrix composed of $k$ blocks $\lambda_l \mathbf{J}$, with $\lambda_l \geq 0$ and

$$\mathbf{J} = \begin{bmatrix} 0 & -1 \\ 1 & 0 \end{bmatrix}.$$

This decomposition leverages the fact that the eigenvalues of $\mathbf{P}$ are purely imaginary and occur in conjugate pairs $\pm i\lambda_l$.

Define the block-diagonal matrix $\mathbf{R}^{\succ} = \mathrm{blockdiag}(\mathbf{J}, \ldots, \mathbf{J}) \in \mathbb{R}^{2k \times 2k}$, and let

$\mathbf{D} = \mathrm{blockdiag}(\sqrt{\lambda_1}\mathbf{I}_2, \ldots, \sqrt{\lambda_k}\mathbf{I}_2) \in \mathbb{R}^{2k \times 2k}$, where $\mathbf{I}_2$ is the $2 \times 2$ identity matrix.

Observe that $\mathbf{\Lambda} = \mathbf{D}\mathbf{R}^{\succ}\mathbf{D}$.

Set $\mathbf{V} = \mathbf{U}\mathbf{D}$. Then,

$$\mathbf{P} = \mathbf{U}\mathbf{\Lambda}\mathbf{U}^{\top} = \mathbf{U}\mathbf{D}\mathbf{R}^{\succ}\mathbf{D}\mathbf{U}^{\top} = \mathbf{V}\mathbf{R}^{\succ}\mathbf{V}^{\top}.$$

Therefore,

$$P_{ij} = \mathbf{v}_i^{\top}\mathbf{R}^{\succ}\mathbf{v}_j, \quad \forall i, j,$$

where $\mathbf{v}_i$ is the $i$-th row of $\mathbf{V}$.

This construction shows that any real skew-symmetric matrix $\mathbf{P}$ can be represented in terms of embeddings $\{\mathbf{v}_i\}$ and the asymmetric operator $\mathbf{R}^{\succ}$, confirming the full expressiveness of our preference representation model. $\qquad\square$

### C.4. Proof of Theorem 5.1.

*Proof.* The proof follows the logic of standard multiplicative weights update analysis, adapted for our preference score objective. First, since the preference score $s$ is bounded in $[-\rho, \rho]$, we can normalize it to $[0, 1]$ by the transformation:

$$\widetilde{s}(\mathbf{y} \succ \mathbf{y}' \mid \mathbf{x}) = \frac{s(\mathbf{y} \succ \mathbf{y}' \mid \mathbf{x})}{2\rho} + \frac{1}{2}$$

By Theorem 1 in (Freund & Schapire, 1999), for any sequence of mixed policies $\mu_1, \mu_2, \ldots, \mu_T$, the sequence of policies $\pi_1, \pi_2, \ldots, \pi_T$ produced by GPO satisfies:

$$\sum_{t=1}^{T} \widetilde{s}(\pi_t \prec \mu_t) \leq \min_{\pi} \left[ \frac{\eta}{1 - e^{-\eta}} \sum_{t=1}^{T} \widetilde{s}(\pi \prec \mu_t) + \frac{\mathrm{KL}(\pi \| \pi_0)}{1 - e^{-\eta}} \right],$$

where $\eta = 1/\beta$ is the learning rate parameter in the MWU view (related to $\beta$ in our GPO loss Eq. 5.3). The preference score $s$ is assumed bounded, which holds in our implementation due to L2 normalization of embeddings and bounded eigenvalues (Section 4.2).

Setting $\mu_t = \pi_t$, note that $\widetilde{s}(\pi_t \prec \pi_t) = \frac{1}{2}$ due to the normalization and symmetry. Thus:

$$\frac{T}{2} \leq \min_{\pi} \left[ \frac{\eta T}{1 - e^{-\eta}} \widetilde{s}(\pi \prec \bar{\pi}_T) + \frac{\mathrm{KL}(\pi \| \pi_0)}{1 - e^{-\eta}} \right]$$

where $\bar{\pi}_T = \frac{1}{T} \sum_{t=1}^{T} \pi_t$ is the mixture policy.

Rearranging terms:

$$\frac{1 - e^{-\eta}}{2\eta} \leq \min_{\pi} \left[ \widetilde{s}(\pi \prec \bar{\pi}_T) + \frac{\mathrm{KL}(\pi \| \pi_0)}{\eta T} \right]$$

Since $\pi_0$ is an autoregressive model with finite vocabulary support, $|\log \pi_0(\cdot)|$ is bounded from above. Thus:

$$\mathrm{KL}(\pi \| \pi_0) \leq \| \log \pi_0(\cdot) \|_{\infty}$$

Setting $\eta = \frac{\| \log \pi_0(\cdot) \|_{\infty}}{\sqrt{T}}$ and using Taylor expansion $\frac{1 - e^{-\eta}}{2\eta} = \frac{1}{2} - \frac{\eta}{4} + O(\eta^2)$:

$$\frac{1}{2} - \frac{\| \log \pi_0(\cdot) \|_{\infty}}{4\sqrt{T}} + O(T^{-1}) \leq \min_{\pi} [\widetilde{s}(\pi \prec \bar{\pi}_T)] + \sqrt{\frac{\| \log \pi_0(\cdot) \|_{\infty}}{T}}$$

Converting back to the original preference score scale:

$$\min_{\pi} [s(\pi \prec \bar{\pi}_T)] \geq -\frac{\rho}{2} - O\left(\frac{\rho}{\sqrt{T}}\right)$$

By symmetry:

$$\max_{\pi} [s(\pi \succ \bar{\pi}_T)] \leq \frac{\rho}{2} + O\left(\frac{\rho}{\sqrt{T}}\right)$$

Therefore, the duality gap is:

$$\max_{\pi} s(\pi \succ \bar{\pi}_T) - \min_{\pi} s(\pi \prec \bar{\pi}_T)$$
$$= \max_{\pi} s(\pi \succ \bar{\pi}_T) - \min_{\pi} s(\pi \prec \bar{\pi}_T)$$
$$= O\left(\frac{1}{\sqrt{T}}\right)$$

$\square$

**Proof of Theorem A.2.**

*Proof.* We aim to represent any real skew-symmetric matrix $\mathbf{P} \in \mathbb{R}^{k \times k}$ using the imaginary parts of inner products of complex vectors.

For each $i = 1, \ldots, k$, define the complex vector $\mathbf{v}_i = \mathbf{a}_i + i\,\mathbf{b}_i$, where $\mathbf{a}_i, \mathbf{b}_i \in \mathbb{R}^k$. Let $\mathbf{a}_i = \mathbf{e}_i$, the $i$-th standard basis vector in $\mathbb{R}^k$, and set

$$\mathbf{b}_i = \frac{1}{2} \sum_{j=1}^{k} P_{ij} \mathbf{e}_j.$$

This implies that the $j$-th component of $\mathbf{b}_i$ is $(\mathbf{b}_i)_j = \frac{1}{2} P_{ij}$.

The Hermitian inner product of $\mathbf{v}_i$ and $\mathbf{v}_j$ is

$$\langle \mathbf{v}_i, \mathbf{v}_j \rangle = (\mathbf{a}_i^\top - i\,\mathbf{b}_i^\top)(\mathbf{a}_j + i\,\mathbf{b}_j) = \mathbf{a}_i^\top \mathbf{a}_j + \mathbf{b}_i^\top \mathbf{b}_j + i\,(\mathbf{b}_i^\top \mathbf{a}_j - \mathbf{a}_i^\top \mathbf{b}_j).$$

Therefore,

$$\mathrm{Im}\left(\langle \mathbf{v}_i, \mathbf{v}_j \rangle\right) = \mathbf{b}_i^\top \mathbf{a}_j - \mathbf{a}_i^\top \mathbf{b}_j.$$

Compute $\mathbf{b}_i^\top \mathbf{a}_j$ and $\mathbf{a}_i^\top \mathbf{b}_j$:

$$\mathbf{b}_i^\top \mathbf{a}_j = (\mathbf{b}_i)_j = \frac{1}{2} P_{ij},$$
$$\mathbf{a}_i^\top \mathbf{b}_j = (\mathbf{b}_j)_i = \frac{1}{2} P_{ji} = -\frac{1}{2} P_{ij},$$

since $P_{ji} = -P_{ij}$ due to skew-symmetry.

Thus,

$$\mathrm{Im}\left(\langle \mathbf{v}_i, \mathbf{v}_j \rangle\right) = \frac{1}{2} P_{ij} - \left(-\frac{1}{2} P_{ij}\right) = P_{ij}.$$

Therefore, we have constructed complex vectors $\mathbf{v}_i$ such that

$$P_{ij} = \mathrm{Im}\left(\langle \mathbf{v}_i, \mathbf{v}_j \rangle\right), \quad \forall\,i, j.$$

This completes the proof. $\square$

# D. More Related Work

**Intransitivity in Game Theory.** The symmetric zero-sum game and its intransitivity have also been frequently studied in the context of game theory. Balduzzi et al. (2018) explored game decompositions into transitive and cyclic components, proposing Nash averaging for agent evaluation. Balduzzi et al. (2019) generalized the results from matrix games to functional-form games and proposed algorithms for diverse agent populations. Czarnecki et al. (2020) investigated the geometrical properties of real-world games (e.g., Tic-Tac-Toe, Go, StarCraft II) and proposed that real-world games have a "spinning top" geometry, with a strong transitive dimension and gradually diminishing non-transitive cyclic dimensions. Very recently, Bertrand et al. (2023) examined the limitations of the Elo rating system and proposed an alternative "disc decomposition" method that can better handle both transitive and cyclic game dynamics.

**Representation Learning and Embedding.** Representation learning involves learning transformations of data that make it easier to extract useful information (Bengio et al., 2013). Techniques like word embeddings (Mikolov et al., 2013) and contrastive learning (Chen et al., 2020; Radford et al., 2021) learn powerful representations for various tasks. While these methods capture semantics or similarities, their direct application to modeling the directed, potentially asymmetric, and intransitive nature of human preferences in RLHF has been less explored. Our GPM specifically introduces a structured embedding space with a skew-symmetric operator tailored to capture these preference characteristics, offering a novel application of representation learning principles to the alignment problem.

**Alternative Preference/Reward Models.** Beyond the standard BT model, other approaches exist. Pairwise preference models (Section 3.3) often concatenate pairs, potentially suffering from order effects and quadratic complexity in the number of responses per prompt. Multi-dimensional reward models like ArmoRM (Wang et al., 2024a) decompose reward into interpretable dimensions (e.g., helpfulness, harmlessness). While offering interpretability, they typically require multi-dimensional annotations and, being based on summing scalar rewards, may still struggle to represent complex intransitive structures compared to GPM's more general formulation (Theorem 4.4). GPM aims for expressiveness and automatic discovery of preference dimensions from standard pairwise labels.

**Preference Optimization Methods.** Recent work has explored various algorithms beyond PPO for RLHF. DPO (Rafailov et al., 2024) directly optimizes the policy using a derived reward. IPO (Azar et al., 2023) and related works (Mitchell, 2023; Liang et al., 2024; Furuta et al., 2024) focus on robustness or handling noisy/soft preferences within the DPO framework. For instance, Mitchell (2023) analyzed the relationship between DPO and IPO under noisy preferences, while Liang et al. (2024) and Furuta et al. (2024) proposed robust preference optimization methods to handle noise and soft labels, respectively. Nash-based methods (Munos et al., 2023; Swamy et al., 2024; Rosset et al., 2024; Wu et al., 2024b) explicitly consider the game-theoretic aspect of finding optimal policies. Our GPO (Section 5) builds upon these ideas but uses the preference score $s$ directly, offering an alternative optimization target compared to win rates or implicit rewards. GPM's preference scores can potentially be integrated into many of these existing optimization frameworks.

# E. More on Experiments

**Cyclic Preference Dataset.** We constructed a dataset by inducing cyclic preferences from the Ultrafeedback dataset (Cui et al., 2024). The dataset includes responses evaluated across four key metrics: *instruction following*, *honesty*, *truthfulness*, and *helpfulness*. We created preference cycles such as: `instruction following` $\succ$ `honesty` $\succ$ `truthfulness` $\succ$ `helpfulness` $\succ$ `instruction following`, ensuring the presence of intransitive cycles. We further generated four sub-datasets by omitting one metric from each cycle, resulting in 4 different datasets with 216 to 363 instances.

We construct the Cyclic Preference Dataset using the following steps: 1. Filter Ultrafeedback for prompts with at least 3 responses where multi-aspect ratings (helpfulness, honesty, etc.) are available.

2. For a set of 3 responses (A, B, C) to the same prompt, check if ratings imply a cycle based on chosen criteria. Example: If criterion 1 is Honesty and criterion 2 is Helpfulness, check if Rating(A, Honesty) $\succ$ Rating(B, Honesty), Rating(B, Helpfulness) $\succ$ Rating(C, Helpfulness), and Rating(C, Honesty) $\succ$ Rating(A, Honesty). This implies a potential cycle A $\succ$ B $\succ$ C $\succ$ A across criteria. 3. Define pairwise preferences based on the higher rating for the specific criterion involved in the cycle link (e.g., A $\succ$ B based on Honesty, B $\succ$ C based on Helpfulness, C $\succ$ A based on Honesty). 4. Collect these cyclic triplets (or longer cycles) to form the datasets used in Table 1. Different datasets (Cyclic No. 1-4) were created by focusing on cycles involving different combinations of the four criteria.

## E.1. Additional Ablation Studies

**Ablations on Scale Gate and Embedding head.** We investigate the effects of scale gates and embedding head dimensions, with and without L2 normalization, on model performance. As shown in Table 4, for Gemma-2B-it models, incorporating a scale gate generally enhances GPM performance across various embedding dimensions. L2 normalization on the embedding head output consistently improves models with scale gates. Interestingly, Gemma-2B-it-based models without L2 normalization or scale gates outperform those with L2 normalization but no scale gates. A plausible explanation for this phenomenon is that removing L2 normalization introduces additional degrees of freedom, particularly beneficial for models with smaller parameter spaces and high-dimensional embedding layers. This increased flexibility may allow the model to utilize its limited parametric capacity better, potentially leading to enhanced expressiveness and task-specific adaptability.

*Table 4.* Impact of the embedding head and the scale gate on GPM's performance on RewardBench. Dim. represents the dimension of the embedding head. The highest average scores for each base model are in bold.

| Embedding Type | Dim. | Chat | Chat-Hard | Safety | Reasoning | Average |
|---|---|---|---|---|---|---|
| **Base Model: Gemma-2B-it** | | | | | | |
| w. scale gate w. l2 | 2 | 77.37 | 73.46 | 85.00 | 85.50 | 80.33 |
| w. scale gate w.o. l2 | 2 | 79.33 | 74.34 | 85.14 | 88.41 | **81.80** |
| w. o. scale gate w. l2 | 2 | 78.49 | 71.27 | 85.68 | 86.13 | 80.39 |
| w. o. scale gate w.o. l2 | 2 | 79.05 | 73.46 | 84.86 | 86.56 | 80.98 |
| w. scale gate w. l2 | 4 | 78.77 | 72.59 | 85.44 | 84.82 | 80.43 |
| w. scale gate w.o. l2 | 4 | 80.45 | 72.81 | 84.46 | 87.61 | **81.33** |
| w. o. scale gate w. l2 | 4 | 79.61 | 70.39 | 85.00 | 86.84 | 80.46 |
| w. o. scale gate w.o. l2 | 4 | 80.72 | 73.02 | 83.51 | 86.96 | 81.06 |
| w. scale gate w. l2 | 6 | 79.61 | 75.66 | 85.27 | 88.61 | **82.29** |
| w. scale gate w.o. l2 | 6 | 76.54 | 76.10 | 85.14 | 87.55 | 81.33 |
| w. o. scale gate w. l2 | 6 | 79.61 | 71.05 | 85.81 | 87.74 | 81.05 |
| w. o. scale gate w.o. l2 | 6 | 77.93 | 73.25 | 85.41 | 86.66 | 80.81 |
| w. scale gate w. l2 | 8 | 78.49 | 74.34 | 84.19 | 86.95 | 81.00 |
| w. scale gate w.o. l2 | 8 | 82.40 | 74.78 | 85.54 | 85.47 | **82.05** |
| w. o. scale gate w. l2 | 8 | 77.09 | 72.15 | 86.08 | 85.41 | 80.18 |
| w. o. scale gate w.o. l2 | 8 | 81.28 | 73.25 | 84.59 | 85.90 | 81.26 |
| **Base Model: Llama-3.1-8B-Instruct** | | | | | | |
| w. scale gate w. l2 | 2 | 91.62 | 88.38 | 90.68 | 94.82 | 91.37 |
| w. scale gate w.o. l2 | 2 | 93.85 | 86.84 | 90.68 | 91.60 | 90.74 |
| w. o. scale gate w. l2 | 2 | 92.18 | 86.18 | 91.89 | 94.05 | 91.08 |
| w. o. scale gate w.o. l2 | 2 | 93.30 | 87.94 | 91.22 | 93.55 | **91.50** |
| w. scale gate w. l2 | 4 | 93.30 | 86.18 | 91.22 | 95.69 | **91.60** |
| w. scale gate w.o. l2 | 4 | 94.13 | 86.18 | 89.86 | 90.55 | 90.18 |
| w. o. scale gate w. l2 | 4 | 92.46 | 87.28 | 91.76 | 93.19 | 91.17 |
| w. o. scale gate w.o. l2 | 4 | 93.58 | 86.40 | 90.95 | 95.33 | 91.56 |
| w. scale gate w. l2 | 6 | 91.90 | 87.50 | 91.62 | 96.40 | **91.86** |
| w. scale gate w.o. l2 | 6 | 93.02 | 85.75 | 91.08 | 91.31 | 90.29 |
| w. o. scale gate w. l2 | 6 | 92.18 | 85.53 | 90.81 | 94.20 | 90.68 |
| w. o. scale gate w.o. l2 | 6 | 93.30 | 87.94 | 90.95 | 90.90 | 90.77 |
| w. scale gate w. l2 | 8 | 93.58 | 87.50 | 91.08 | 95.44 | **91.90** |
| w. scale gate w.o. l2 | 8 | 93.02 | 87.06 | 90.81 | 92.20 | 90.77 |
| w. o. scale gate w. l2 | 8 | 91.90 | 86.62 | 91.22 | 92.63 | 90.59 |
| w. o. scale gate w.o. l2 | 8 | 93.02 | 87.72 | 90.68 | 90.16 | 90.39 |

## E.2. Additional Experimental Results

**More Results on Language Model Alignment.** We further conduct additional evaluations of our fine-tuned models using various benchmarks. AlpacaEval 2.0 evaluation results are listed in Table 5, using GPT-4o-mini as evaluators. For MT-Bench, we used the default mode to let GPT-4 grade and give a score to the model's answer, and the MT-Bench scores of aligned models are presented in Table 6.

*Table 5.* AlpacaEval 2.0 evaluation results. Base model: Llama3-8B-it, Evaluator: GPT-4o-mini. The results are grouped by the size and type of the RM or PM, and the number of iterations. Bold entries indicate that GPM outperforms BT RM under the same training settings.

| Size | Type | Iter | SPPO | | | GPO | | |
|---|---|---|---|---|---|---|---|---|
| | | | LC. WR | WR | Avg. Len | LC. WR | WR | Avg. Len |
| | | base | 23.07 | 32.26 | 1959 | 23.07 | 32.26 | 1959 |
| 2B | BT RM | 1 | 48.84 | 46.09 | 1939 | 53.15 | 49.94 | 1929 |
| | | 2 | 59.77 | 58.41 | 2032 | 66.19 | 64.88 | 2049 |
| | | 3 | 66.81 | 67.14 | 2136 | 71.75 | 71.68 | 2151 |
| | GPM | 1 | 48.09 | **49.15 (+3.06)** | 2066 | 55.66 | **57.12 (+7.18)** | 2102 |
| | | 2 | 56.63 | **63.53 (+5.12)** | 2301 | 61.11 | **67.78 (+2.90)** | 2343 |
| | | 3 | 60.77 | **70.91 (+3.77)** | 2498 | 64.52 | **74.78 (+3.10)** | 2582 |
| 8B | BT RM | 1 | 45.24 | 36.95 | 1740 | 49.77 | 40.26 | 1702 |
| | | 2 | 56.24 | 50.36 | 1868 | 60.75 | 56.30 | 1933 |
| | | 3 | 63.71 | 58.38 | 1948 | 62.63 | 59.17 | 1969 |
| | GPM | 1 | 46.84 | **41.42 (+4.47)** | 1861 | 53.12 | **46.64 (+6.38)** | 1850 |
| | | 2 | 58.03 | **56.07 (+5.71)** | 2029 | 59.86 | **60.37 (+4.07)** | 2115 |
| | | 3 | 61.64 | **63.42 (+5.04)** | 2385 | 62.51 | **67.48 (+8.31)** | 3249 |

*Table 6.* MT-Bench evaluation results. Base model: Llama3-8B-it, Evaluator: GPT-4. Bold entries indicate that GPM outperforms BT RM under the same training settings.

| Size | Type | Iter | SPPO | | | GPO | | |
|---|---|---|---|---|---|---|---|---|
| | | | 1st | 2nd | Avg. | 1st | 2nd | Avg. |
| | | base | 8.31 | 7.77 | 8.03 | 8.31 | 7.77 | 8.03 |
| 2B | BT RM | 1 | 8.42 | 7.57 | 8.00 | 8.33 | 7.85 | 8.09 |
| | | 2 | 8.20 | 7.73 | 7.96 | 8.30 | 7.66 | 7.98 |
| | | 3 | 8.44 | 7.66 | 8.05 | 8.41 | 8.09 | 8.25 |
| | GPM | 1 | 8.23 | 7.65 | 7.94 | **8.70** | **7.95** | **8.33** |
| | | 2 | **8.53** | **8.24** | **8.38** | **8.69** | **8.01** | **8.35** |
| | | 3 | 8.39 | **7.84** | **8.12** | **8.48** | 7.76 | 8.12 |
| 8B | BT RM | 1 | 8.44 | 8.10 | 8.27 | 8.41 | 7.85 | 8.13 |
| | | 2 | 8.75 | 7.85 | 8.30 | 8.73 | 7.83 | 8.28 |
| | | 3 | 8.34 | 7.99 | 8.17 | 8.68 | 7.83 | 8.26 |
| | GPM | 1 | 8.43 | 7.94 | 8.18 | 8.29 | **7.90** | 8.10 |
| | | 2 | 8.51 | **8.05** | 8.28 | 8.26 | **7.99** | 8.13 |
| | | 3 | 8.47 | 7.76 | 8.12 | 7.57 | 7.51 | 7.54 |

**Discussion on Length Control (LC. WR).** As observed in Section 6.3 (Table 3, 5), GPM-aligned models often produce longer responses than BT-aligned models. The standard AlpacaEval Win Rate (WR) doesn't penalize length, whereas the Length-Controlled Win Rate (LC. WR) is designed to mitigate length bias, potentially penalizing models that win primarily by being more verbose. GPM's multi-dimensional embeddings might better capture the value of comprehensive answers that address multiple facets of a prompt (e.g., being helpful, detailed, and stylistically appropriate simultaneously), leading to higher preference scores and thus longer generated responses under GPO. While beneficial for overall quality perceived by the preference model, this can negatively impact the LC. WR metric. For applications where conciseness is highly valued or length bias is a major concern, specific techniques might be needed. One approach is length normalization within the optimization objective.

**Length-Normalized GPO (LN-GPO) Results.** We explored a variant, Length-Normalized GPO (LN-GPO), which modifies the GPO loss (Eq. 5.3) to normalize the policy update by response length, similar in spirit to Meng et al. (2024). The objective becomes:

$$\mathcal{L}_{\text{LN-GPO}}(\boldsymbol{\theta}) = \mathbb{E}_{\mathbf{x}\sim\mathcal{X},\mathbf{y}\sim\pi_{\boldsymbol{\theta}_t}(\cdot|\mathbf{x})}\left[\left(\frac{1}{|\mathbf{y}|^\gamma}\log\left(\frac{\pi_{\boldsymbol{\theta}}(\mathbf{y}\mid\mathbf{x})}{\pi_{\boldsymbol{\theta}_t}(\mathbf{y}\mid\mathbf{x})}\right) - \frac{1}{\beta}\left(\widehat{s}(\mathbf{y}\succ\pi_{\boldsymbol{\theta}_t}\mid\mathbf{x}) - \log Z_{\pi_{\boldsymbol{\theta}_t}}(\mathbf{x})\right)\right)^2\right].$$

Here, $|\mathbf{y}|$ is the length of the response $\mathbf{y}$, and $\gamma$ is a hyperparameter (typically 1). Initial results using LN-GPO with $\gamma = 1$

are shown in Table 7. We observe that LN-GPO with GPM (2B) achieves a slightly higher LC. WR (45.55%) compared to LN-GPO with BT RM (45.51%), while producing longer responses (Avg. Length 2112 vs 1951). This suggests length normalization can help, although further tuning and investigation are needed to fully balance performance and length control with GPM.

*Table 7.* AlpacaEval 2.0 evaluation results with LN-GPO. Base model: Llama3-8B-it. Evaluator: gpt-4o-mini.

| Model | Win Rate (%) | Avg. Length | LC. WR (%) |
|---|---|---|---|
| `LN-GPO-Llama-3-8B-Instruct-Iter1_gp_2b` | **48.31** | 2112 | **45.55** |
| `LN-GPO-Llama-3-8B-Instruct-Iter1_bt_2b` | 43.38 | 1951 | 45.51 |

### E.3. Implementation Details

**Details on Training Setup.** Our experiments on RewardBench and Cyclic Preference Dataset were implemented using the HuggingFace Transformers library (Wolf et al., 2020) and the OpenRLHF framework (Hu et al., 2024). For reward model training on Skywork Reward Data Collection, we employed the following settings (in Table 8):

- **Gemma-2B-it:** Trained with a learning rate of $2 \times 10^{-6}$.
- **Llama-3.1-8B-Instruct:** Trained with a learning rate of $2 \times 10^{-6}$.
- **Gemma-2-9B-it:** Trained with a learning rate of $2 \times 10^{-6}$.
- **Training Configuration:** Both models were trained for two epochs with a global batch size of 32. We used a cosine learning rate scheduler with a warm-up ratio of 0.03. Input sequences were truncated to a maximum length of 2048 tokens.
- **Hyperparameters:** For our general preference embedding model (GPM), we set $\beta = 0.1$, determined via hyperparameter tuning on a validation set.
- **Hardware:** All experiments were conducted on machines equipped with NVIDIA A800 80GB GPUs, utilizing 8 GPUs per experiment.

For cyclic preference experiments, the training settings are as follows, except for the parameters specified below; all other experimental parameters remain consistent with experiments on RewardBench (in Table 9):

- **Gemma-2B-it:** Trained with a learning rate of $1 \times 10^{-6}$.
- **Training Configuration:** Models were trained for 50 epochs with a global batch size of 1.
- **Hardware:** Experiments were conducted on machines equipped with NVIDIA A800 80GB GPUs, utilizing a single GPU per experiment.

**Details on Evaluation Dataset RewardBench.** RewardBench is divided into four core sections:

- **Chat:** Evaluates the ability to differentiate between thorough and correct responses in open-ended conversations, using data from AlpacaEval (Li et al., 2023) and MT Bench (Zheng et al., 2023).
- **Chat-Hard:** Tests the handling of trick questions and subtle instruction differences, using adversarial examples from MT Bench and LLMBar (Zeng et al., 2024).
- **Safety:** Assesses the capacity to refuse harmful content appropriately, using data from XSTest (Röttger et al., 2024), Do-Not-Answer (Wang et al., 2024b), and a custom AI2 dataset.
- **Reasoning:** Measures code generation and reasoning abilities, with prompts from HumanEvalPack (Muennighoff et al., 2023) and PRM800k (Lightman et al., 2023).

## F. Examples on Ultrafeedback Dataset

To illustrate that cyclic preferences can occur even without changing the criteria, we present several examples from the Ultrafeedback dataset (Cui et al., 2024).

### F.1. Example 1

**Prompt:** "Provide a detailed history and significance of the Mississippi Arts Center in Jackson"

*Table 8.* Implementation details for experiments on RewardBench.

| General Settings | |
|---|---|
| Base models | Gemma-2b-it and Llama3.1-8B-Instruct |
| Batch size | 32 |
| Quantization for training | bf16 |
| Learning Rate | $2 \times 10^{-6}$ |
| Learning Rate Scheduler | cosine |
| Warmup Ratio | 0.03 |
| Max training epochs | 2 |
| Gradient accumulation step | 1 |
| Max input length | 2048 |
| Zero stage | 3 |
| Flash attention enabled | True |
| **General Preference Model** | |
| $\beta$ for loss function | 0.1 |

*Table 9.* Implementation details for experiments on Cyclic Preference Dataset.

| General Settings | |
|---|---|
| Base models | Gemma-2b-it |
| Batch size | 1 |
| Quantization for training | bf16 |
| Learning Rate | $1 \times 10^{-6}$ |
| Learning Rate Scheduler | cosine |
| Warmup Ratio | 0.03 |
| Max training epochs | 50 |
| Gradient accumulation step | 1 |
| Max input length | 2048 |
| Zero stage | 3 |
| Flash attention enabled | True |
| **General Preference Model** | |
| $\beta$ for loss function | 0.1 |

We have three responses within the Ultrafeedback dataset:

RESPONSE A:

"*The Mississippi Arts Center, located in Jackson, Mississippi, is a vital cultural institution that has played a significant role in the state's arts scene for decades. It is a facility of the Mississippi Department of Archives and History and is home to the Mississippi Arts Commission, Mississippi Craft Center, Mississippi History Museum, and the Eudora Welty House Museum.*

*The Mississippi Arts Center was established in 1968 as the Mississippi Arts Commission. Its primary purpose is to support and develop the arts in Mississippi through a variety of programs and initiatives. Over the years, it has provided grants to arts organizations, schools, and individual artists; managed a state artist roster; developed arts education programs; and promoted cultural tourism in Mississippi.*

*The physical Mississippi Arts Center building opened in 2004 and houses various facilities and spaces for art exhibitions, education, and programs. The Mississippi Craft Center is a featured part of the arts center, displaying and selling the works of local artisans and craftsmen. It also offers educational programs, classes, and workshops for both children and adults.*

*The Mississippi History Museum is another important component of the arts center. The museum explores the state's history*

*from prehistoric times up to the present day, featuring exhibits on Mississippi's Native American culture, the Civil War, and the Civil Rights Movement.*

*In addition to these facilities, the Mississippi Arts Center is also home to the Eudora Welty House Museum, which is dedicated to the life and work of Eudora Welty, a renowned short-story writer and photographer from Mississippi. The museum features tours of her home, which has been restored to its original condition.*

*The Mississippi Arts Center in Jackson holds significant importance as a vital cultural institution and resource for both the local community and the state of Mississippi. It plays a crucial role in promoting the arts, preserving history, and providing educational opportunities related to Mississippi's rich cultural heritage. As a result, it has become an essential part of Jackson's cultural landscape and a must-visit for those interested in exploring Mississippi's unique history and arts scene. "*

Response A provides extensive information about the Mississippi Arts Center but mixes in details about other institutions like the Mississippi Arts Commission and the Mississippi History Museum, leading to partial inaccuracies.

**Ratings of Response A.**

- **Instruction Following:** 2 (Partially addresses the instruction but includes unrelated institutions)

- **Honesty:** 5 (Confident and detailed)

- **Truthfulness:** 3 (Some inaccuracies due to mixing institutions)

- **Helpfulness:** 4 (Informative but somewhat confusing)

RESPONSE B:

" *The Mississippi Arts Center (MAC) is a state-funded, non-profit art museum located in Jackson, Mississippi. It is the largest art museum in the state, and its mission is to "inspire creativity, enrich lives, and connect communities through the arts."*

*The MAC was founded in 1938 as the Mississippi Fine Arts Commission. It was originally housed in the Old Capitol building in Jackson. In 1976, the MAC moved to its current location in the former Woolworth's building on Capitol Street.*

*The MAC's collection includes over 15,000 works of art, including paintings, sculptures, drawings, prints, photographs, and decorative arts. The collection represents a wide range of artists from Mississippi and the American South.*

*The MAC also offers a variety of educational programs, including art classes, workshops, lectures, and tours. The MAC's Education Department serves over 100,000 people each year through its programs and outreach activities.*

*The MAC is a significant cultural institution in Mississippi. It is a place where people can come to learn about the arts, create art, and experience the power of art to enrich lives.*

*Here are some additional facts about the Mississippi Arts Center: - The MAC's building was originally a Woolworth's department store. It was designed by architect James Gamble Rogers, who also designed the Woolworth's buildings in New York City and Washington, D.C. - The MAC's collection includes works by some of the most important artists of the American South, including William Eggleston, Walker Evans, and Eudora Welty. - The MAC has hosted a number of important exhibitions, including the first solo museum exhibition of the work of painter Kerry James Marshall. - The MAC is a member of the American Alliance of Museums. - The MAC is accredited by the American Alliance of Museums. "*

Response B offers detailed facts about the Mississippi Arts Center but includes significant errors, such as incorrect founding year and location, despite being confident.

**Ratings of Response B.**

- **Instruction Following:** 3 (Addresses some aspects but makes errors)

- **Honesty:** 2 (Confident but includes significant mistakes)

- **Truthfulness:** 5 (Avoids fabrication but contains factual errors)

- **Helpfulness:** 3 (Detailed but confusing key institutions)

RESPONSE C:

" *The Mississippi Arts Center, also known as the Mississippi Museum of Art in Jackson, has a rich history and significance in the state's cultural and artistic landscape. Founded in 1911 as the Mississippi Art Association, its primary mission was to promote the understanding, appreciation, and enjoyment of the visual arts in Mississippi.*

*In the beginning, the association held exhibitions in various locations across the state. However, in 1978, the Mississippi Museum of Art at the Mississippi Agency building in Downtown Jackson was established, providing a permanent home where the association could showcase its growing collection. Over time, the museum has evolved and expanded.*

*In 2007, the museum relocated to a new, state-of-the-art facility at 380 South Lamar Street in Downtown Jackson. This move allowed for the enlargement of galleries, versatile exhibition spaces, and the incorporation of educational programs and events.* "

Response C delivers a comprehensive history of the Mississippi Museum of Art, mistakenly equating it with the Mississippi Arts Center, but avoids fabrication.

**Ratings of Response C.**

- **Instruction Following:** 5 (Comprehensive but misinterprets the prompt)

- **Honesty:** 4 (Accurate information but about a different institution)

- **Truthfulness:** 3 (Misunderstands the instruction)

- **Helpfulness:** 5 (Thorough and informative)

**Analysis.** Response A preferred over Response B: A is more accurate than B, which contains significant errors; Response B preferred over Response C: Some evaluators prefer B over C because B discusses the correct institution (despite errors), while C focuses on a different one. Response C preferred over Response A: Others prefer C over A because C provides a thorough and helpful response, even if it's about a different institution, whereas A includes confusing mixed references. This forms a cycle: $A \succ B \succ C \succ A$, even though the question and evaluation criteria remain the same.

**F.2. Example 2**

**Prompt:** "A whole school, a class-based cooking competition that runs throughout the week. Which class will win this year? Winners announced on Monday 10th July in assembly. Based on the passage above, Could you provide some more details about the cooking competition, such as the criteria for judging and the types of dishes being prepared?"

We have three responses within the Ultrafeedback dataset:

RESPONSE A:

"*The Mississippi Arts Center, located in Jackson, Mississippi, is a vital cultural institution that has played a significant role in the state's arts scene for decades. It is a facility of the Mississippi Department of Archives and History and is home to the Mississippi Arts Commission, Mississippi Craft Center, Mississippi History Museum, and the Eudora Welty House Museum.*

*Sure, here are some additional details about the school's cooking competition: The cooking competition is divided into three stages:*

*Stage 1: Prep Day - During this stage, students are given a list of ingredients and a set period of time to prepare their dish. The goal of this stage is to encourage creativity and problem-solving skills as students work with limited time and resources.*

*Stage 2: Cook-off Day - In this stage, students prepare their dishes in front of a panel of judges. The judges will evaluate the dishes based on a set of criteria, such as taste, presentation, and creativity.*

*Stage 3: Judging - After the cook-off, the judges deliberate and decide on the winning class. The winning class will be announced during a school assembly and will receive a prize.*

*The dishes prepared in the competition range from simple to complex, and each class is required to prepare a different dish. The criteria for judging the dishes include:*

*Taste: How well does the dish taste? Is it flavorful and balanced?*

*Presentation: How well is the dish presented? Is it appealing to the eye?*

*Creativity: Was the dish creatively prepared? Did the student use unusual ingredients or techniques?*

*Overall Impression: How well did the student execute the dish? Did they meet the brief and deliver a high-quality result?*

*I hope this offers a more comprehensive view of the school's cooking competition.*"

Response A gives a detailed breakdown of the competition stages and judging criteria but introduces a three-stage format not mentioned in the original prompt.

**Ratings of Response A.**

- **Instruction Following:** 5 - Provides detailed information about the competition's structure, judging criteria, and dish preparation

- **Honesty:** 4 - Mostly correct, but some stages (e.g., Prep Day) may have been inferred rather than drawn from the given passage.

- **Truthfulness:** 3 – Introduces a three-stage competition format not mentioned in the original passage.

- **Helpfulness:** 4 – Highly informative, but lacks details about the specific types of dishes being prepared.

RESPONSE B:

"*Absolutely! It sounds like an exciting and fun-filled school cooking competition, with classes participating throughout the week. Here's some additional information based on the passage provided:*

*Criteria for Judging: Taste: The dishes should be flavorful, delicious, and appealing to the palate. Presentation: The dishes should be visually appealing, with attention given to plating and garnishes.*

*Creativity: Participants should strive to create unique, original dishes or add a twist to traditional recipes.*

*Teamwork: The class's ability to work cohesively and efficiently will also be taken into consideration.*

*Hygiene & Safety: Proper food handling practices and cleanliness will be of importance during the competition.*

*Types of Dishes Being Prepared:*

*Since the specific types of dishes are not mentioned in the passage, here are some suggestions for categories that could be included in the competition:*

*Appetizers: Finger foods, dips, or small bites showcasing culinary skills. Main Courses: Protein-based dishes with sides, vegetarian options, or one-pot meals. International Cuisine: Dishes inspired by the flavors and techniques of different regions.*

*Desserts: Classic cakes, cookies, pastries, or innovative treats.* Budget-Friendly: Nutritious and tasty dishes with minimal costs, teaching budgeting skills. *Winners of the competition will be announced on Monday, the 10th of July during the assembly. Classes will eagerly await the results, and the winning class will earn the title of "School Cooking Champions" for the year. This competition is sure to be an engaging platform for students to learn new skills, make friends, and have a blast while cooking. Bon appétit!* "

Response B offers speculative details about the competition, including possible dish categories and additional judging criteria not specified in the prompt.

**Ratings of Response B.**

- **Instruction Following:** 4 – Covers judging criteria and possible dish types but includes speculative details not mentioned in the passage.

- **Honesty:** 5 – Confident and mostly correct in providing details.

- **Truthfulness:** 4 – Slight hallucination about judging criteria (e.g., teamwork and hygiene) and dish categories.

- **Helpfulness:** 5 – Informative and engaging, with a well-rounded discussion of potential competition elements.

RESPONSE C:

" *Sure, here are some more details about the cooking competition: The competition is open to all students in the school.*

*Each class will be assigned a different day of the week to cook and present their dish.*

*The criteria for judging will be taste, creativity, and presentation. The winning class will be announced on Monday, 10th July, in the assembly.*

*The types of dishes being prepared are wide-ranging, and students are encouraged to be creative. Some popular dishes in past competitions have included:*

*Pizza*

*Pasta*

*Burgers*

*Sandwiches*

*Cakes*

*Cookies*

*The cooking competition is a great way for students to learn about different cuisines and cultures, as well as to develop their cooking skills. It is also a lot of fun!* "

Response C sticks closely to the prompt, providing clear judging criteria and examples of dishes, but lacks depth compared to the other responses.

**Ratings of Response C.**

- **Instruction Following:** 4 – Provides clear judging criteria and dish examples but lacks depth in detailing the competition format.

- **Honesty:** 4 – Mostly correct but assumes information (e.g., students being encouraged to be creative).

- **Truthfulness:** 5 – Free of hallucinations and accurately expands on possible competition elements.

- **Helpfulness:** 5 (Thorough and informative)

**Analysis.** A preferred over B: Some evaluators might prefer A over B because it provides a structured and detailed response, despite introducing unconfirmed elements; B preferred over C: Others might prefer B over C due to its engaging style and comprehensive coverage, even if some details are speculative; C preferred over A: Meanwhile, some may prefer C over A because it sticks closer to the information provided, avoiding potential inaccuracies introduced by A. This results in a preference cycle: $A \succ B \succ C \succ A$.

