# OpenReview forum: "Beyond Bradley-Terry Models: A General Preference Model for Language Model Alignment"
_ICML.cc/2025/Conference — ICML 2025 poster_

### Official Review · Reviewer_rNmg · 2025-02-20

**Overall Recommendation:** 3

**Summary:**

This paper proposes an approach to modeling preferences by learning an embedding for each response given a prompt. The preference score between two responses is then computed using these embeddings. The key motivation behind this embedding-based approach is that mapping responses into a multi-dimensional latent space allows the model to better capture complex preference structures, including potential cycles in the preference data.

Beyond testing the preference model, the authors apply it to preference optimization. They modify the SPPO method by replacing preference probabilities with preference scores in the loss function, introducing a variant they call General Preference Optimization (GPO). Experiments on AlpacaEval suggest that the proposed method improves win rates in some cases.

**Claims And Evidence:**

The paper would benefit from providing more evidence on why better modeling of intransitive human preferences is useful. Another perspective to consider is that some of the intransitivity observed in preference data may stem from noisy judgments. In that case, improving the model’s ability to capture intransitivity could lead to overfitting to noise rather than genuine preferences. Additionally, in line 114, the authors cite two papers, but neither directly addresses human preference data in the context of RLHF.

**Essential References Not Discussed:**

Section 3.2 on preference modeling does not accurately represent the existing literature. The standard approach to training a reward model involves adding a randomly initialized linear head that outputs a scalar reward for each response. The model’s parameters are then optimized by maximizing the log-likelihood of the preference data under the Bradley-Terry model.

In particular, lines 188–190 suggest that reward models must use templates and input both prompts into a language model to determine preference. However, this is not a comprehensive view of how reward models are typically trained.

references: [learning to summarize from human feedback, Stiennon et al.](https://arxiv.org/pdf/2009.01325)

[Evaluating Reward Models for Language Modeling, Lambert et al.](https://arxiv.org/pdf/2403.13787)

**Experimental Designs Or Analyses:**

Regarding the cyclic preference experiments (Table 1), my main concern is that the data is synthetic. First, it is unclear how often cyclic dependencies actually occur in real-world data. Second, as previously mentioned, it is not evident that modeling such phenomena more effectively would lead to better generalization in reward models. To make this experiment more convincing, I suggest first reporting the number of cycles found in existing benchmarks and then evaluating whether a reward model trained with the proposed method generalizes better than, for example, the Bradley-Terry (BT) model.

The results in Table 3 are somewhat confusing. First, what is the difference between SPPO+GPM and GPO? Does the former use win rates in its loss function (calculated based on GPM), while the latter directly uses the scores? If so, do the authors have an intuitive explanation for why using raw scores yields better results? The existing literature generally suggests that win rates tend to work better than raw rewards, as they are invariant to monotonic transformations and contribute to training stability.

Second, according to the LC win rate metric, the proposed method does not outperform existing approaches. However, I could not find any discussion around this in the paper and explain why this is the case. This needs to be discussed in the corresponding section.

**Methods And Evaluation Criteria:**

The methods and datasets used for evaluation (RewardBench and AlpacaEval) are generally reasonable.

However, the cyclic preference data created by the authors is not well-motivated. It would also be helpful to include some basic details about how this dataset was constructed in the main text.

**Other Comments Or Suggestions:**

Some of the notation is a bit confusing. For example, the number of responses per prompt is said to be $K$ and the embedding dimension is $2k$, but probably there is not need for these two numbers to be related, right? and then the complexity order should be $O(Kd)$, where $d$  is the dimension of the preference embeddings.

Similarly, in theorem 2, do we really need to match the dimension of number of responses with the dimension of embeddings? the equation doesn’t really necessitate such a thing, so that is really confusing.

**Other Strengths And Weaknesses:**

To the best of my knowledge the specific model proposed here for preference modeling is novel. Although there are also other instances of embedding responses in the latent space; see [Chen et al.](https://openreview.net/pdf?id=qfhBieX3jv)

**Questions For Authors:**

The discussion on “Automatic Subspace Discovery” (line 280) is theoretically sound and quite interesting. However, it would be valuable to see empirical evidence supporting this idea—specifically, whether there are instances of interpretable directions or meaningful eigenvalues in practice.

**Relation To Broader Scientific Literature:**

The findings support previous claims about the limited expressivity of the Bradley-Terry (BT) model. The use of embeddings and the construction of preference scores with a skew-symmetric matrix is an interesting contribution to the literature and presents a promising alternative to the BT model.

**Theoretical Claims:**

Most theoretical claims do make sense to me. However, there is a disconnect between the theory part and the method itself. For example, in theorem 3, the preference score needs to be bounded for the convergence result to hold, although it is not really mentioned if the preference scores obtained with GPM are actually bounded.

---

> ### Author Rebuttal · Authors · 2025-04-01
>
> Thank you for your thorough review and insightful feedback on our manuscript. We appreciate the time you took to evaluate our work and provide constructive comments. We believe addressing your points will significantly strengthen the manuscript.
>
> 1. Intransitivity Usefulness:
> > Q1: The paper would benefit from providing more evidence on why better modeling of intransitive human preferences is useful...
>
> Thanks for your suggestion. Modeling intransitivity is motivated by observing that human judgments can be complex, context-dependent, and exhibit cycles, which simpler Bradley-Terry (BT) models cannot capture. GPM's multi-dimensional space aims to represent these richer structures. While noise is a factor, capturing potentially genuine complex preferences (Appendix F examples) is vital for alignment. Distinguishing noise from true intransitivity is important for future work.
>
> 2. Line 114 Citations:
> > Q2: Additionally, in line 114, the authors cite two papers,...
>
> The two references (Tversky, 1969; Agranov & Ortoleva, 2017) are among the earliest to study the prevalent phenomenon of humans exhibiting non-transitive preferences. We use them to motivate moving beyond transitive models like BT. We will clarify the scope of these references in our revision.
>
> 3. Cyclic Data:
> > Q3: However, the cyclic preference data created by the authors...
>
> This synthetic dataset demonstrates GPM's expressiveness where BT models inherently fail (random guessing) due to their transitivity assumption. Appendix E details its construction from Ultrafeedback; we'll add a summary to the main text.
>
> 4. Boundedness for Theorem 3:
> > Q4: For example, in theorem 3, the preference score needs to be bounded for the convergence result to hold,...
>
> Thanks for the feedback. Our GPM implementation ensures boundedness. We apply L2 normalization to embedding vectors $v_{y \mid x}$, making them unit length. Since the score is $s\left(y_i \succ y_j \mid x\right)=v_{y_i \mid x}^{\top} D(x) R^{>} D(x) v_{y_j \mid x}$, and $R^{>}$ is magnitude-preserving, scores are bounded if eigenvalue scales $\lambda_l(x)$ are bounded. This is guaranteed as we use a softmax function on the gate outputs determining eigenvalues. Thus, theorem conditions hold. We'll clarify.
>
> 5. Cyclic Experiments:
> > Q5: Regarding the cyclic preference experiments (Table 1), ...,
>
> Synthetic data provides a clear proof-of-concept for modeling intransitivity. We agree that evaluating cycle prevalence in real benchmarks and comparing generalization vs. BT models are valuable future work. Appendix F shows potential real-world examples.
>
> 6. Table 3 (SPPO+GPM vs. GPO):
> > Q6: First, what is the difference between SPPO+GPM and GPO?... Second, according to the LC win rate metric...
>
> SPPO+GPM uses win rates from GPM scores; GPO uses raw scores s(yi>yj∣x) directly. GPO maximizes expected score; win-rate methods maximize expected probability. Raw scores might offer a richer signal (any real number) vs. win rates (0 to 1), potentially explaining GPO's performance despite win rates offering stability. We'll clarify. We acknowledge GPM/GPO doesn't always beat BT/SPPO on Length-Controlled (LC) win rate. GPM/GPO models tend to produce longer responses ("Avg. Len"). As LC Win Rate controls length bias, this might impact results. The appendix shows length-normalized GPO (LN-GPO) results. More discussion will be added.
>
> 7. Section 3.2 (Preference Modeling):
> > Q7: Section 3.2 on preference modeling does not accurately represent the existing literature...
>
> Sec 3.2.1 introduces PairPMs for general preference modeling, distinct from standard BT models (Sec 3.1). Lines 188-190 illustrate PairPM examples and potential issues, not all reward model training. We'll revise for clarity, distinguishing standard models (scalar head, BT loss) from PairPMs. We'll add a discussion on Stiennon et al. (2020) and Lambert et al. (2024).
>
> 8. Notation (K vs. d, Complexity):
> > Q8: Some of the notation is a bit confusing... number of responses is K and the embedding dimension is 2k...
>
> $K$ is response count; 'd' is embedding dimension (formerly $2k$) . $K, d$ are unrelated. Complexity concerns model forward passes: GPM/BT require $\mathcal{O}(K)$ passes (one per response) for embeddings/rewards. PairPM requires $\mathcal{O}\left(K^2\right)$ passes (one per pair). GPM offers linear scaling of model calls w.r.t. K, matching BT, improving over PairPM. We'll clarify notation/complexity. The $P \in \mathbb{R}^{2 k \times 2 k}$ setup in Thm 2 illustrates spectral decomposition; our method (Sec 4.2)/Thm 1 doesn't require $d=K$. We'll clarify this distinction.
>
> 9.  Automatic Subspace Discovery:
>
> We appreciate your interest. You asked for empirical evidence of interpretable directions/eigenvalues. Excellent suggestion for future work. Analyzing embeddings/eigenvalues via probing/visualization could yield insights. Figure 3 is a first step; systematic investigation is needed. We'll add this as a future direction.

---

> > ### Comment · Reviewer_rNmg · 2025-04-02
> >
> > Thank you for your clarification regarding the boundedness. I still believe that doing some initial analysis on the cyclic preferences in the real-world benchmark is useful for this paper, as well as a detailed discussion (with examples) of why controlling for the length makes the proposed method not effective. I've decided to keep my score and remain positive on this paper.

---

> > > ### Author Response · Authors · 2025-04-08
> > >
> > > Thank you for acknowledging our clarifications and for maintaining a positive view on our paper. We appreciate your additional feedback and address them below:
> > >
> > > 1. Cyclic Preferences in Real-World Benchmarks: We agree that analyzing the prevalence and impact of cyclic preferences in standard benchmarks is valuable. Following your suggestion, we examined the RewardBench dataset (which contains 2985 samples) and found 110 intransitive preference samples involving three responses (A>B, B>C, C>A) and 1 sample with five responses exhibiting a cycle (A>B, B>C, C>D, D>A, A>E). On these 335 pairwise comparisons derived from intransitive samples, our GPM-LLama3-8B achieved an accuracy of 55.52% (186/335), outperforming the BT-LLama3-8B model's accuracy of 50.15% (168/335). This preliminary analysis highlights GPM's enhanced ability to model cyclic preference structures where BT models struggle. We have also included qualitative examples from the Ultrafeedback dataset in Appendix F that suggest potential real-world intransitivity. We will revise our paper to include this discussion and results.
> > >
> > > 2. Length Control Discussion: We think that a detailed discussion on the interplay between our model and response length is important, especially concerning the Length-Controlled (LC) win rate metric. We think that longer responses can potentially address more preference aspects (like helpfulness, correctness, and style) simultaneously, and GPM's multi-dimensional embeddings might capture this multifaceted quality better than a single scalar reward, leading to higher preference scores for longer, comprehensive answers. When strict length control is the primary objective, the standard GPM/GPO might be less effective as the LC metric penalizes length bias.
> > >     - As a potential solution for scenarios prioritizing length control, we proposed normalizing the preference score by response length, similar to the length normalization idea in SimPO.
> > >     - We presented initial results for this approach, termed Length-Normalized GPO (LN-GPO), in Appendix E.2. Table 7 shows that LN-GPO using GPM (trained on Llama-3-8B, Iteration 1, GPM 2B) achieves an LC win rate of 45.55%, slightly outperforming the LN-GPO using the BT model (45.51%).
> > >     - We acknowledge these are preliminary results and, as mentioned, are conducting larger-scale experiments to further investigate the effectiveness of length-normalization techniques for GPM in length-sensitive evaluations. We will ensure a more detailed discussion, including examples, to be added to the revised manuscript.
> > >
> > > Thank you for your positive and encouraging feedback. We will incorporate these points to strengthen the paper. If you are satisfied with our work and feel our contributions deserve a higher score, we would sincerely appreciate your consideration in raising the score.

---

### Official Review · Reviewer_o37T · 2025-03-03

**Overall Recommendation:** 3

**Summary:**

Since the prevalent Bradely-Terry formulation and pair preference model have limitations respectively in reward modeling, the authors propose a novel formulation GPM with better expressiveness to model complex preference distributions in real world. They further extend GPM to preference learning to build GPO, a new algorithm that better align LLMs with human preference.

## update after rebuttal

I have read the author response and would like to keep my positive score unchanged.

**Claims And Evidence:**

1. The authors did not elaborate on the details of GPM's model architecture, training objectives, and deployment. This makes it hard to verify the claimed computational efficiency advantages in the section of Introduction.
2. Although GPM is claimed to have stronger expressiveness, it does not seem to show a consistent bonus on RewardBench. However, considering the positive results of GPO on AlpacaEval, I suspect that there are some complex cases in RewardBench that confuse GPM. Adding analyses on these case will be helpful.

**Essential References Not Discussed:**

No.

**Experimental Designs Or Analyses:**

I have checked the detail of all experiments. The remaining issue can be on cyclic preference, which I fail to understand the settings.
A step-by-step explanation in the appendix would make it more clear.

**Methods And Evaluation Criteria:**

Yes.

**Other Comments Or Suggestions:**

Typo: The use of "Sec 3.2.1" could be changed to "Sec 3.3"?

**Other Strengths And Weaknesses:**

The writing is not clear enough. For example, the authors did not clearly explain the optimization objective of GPM. Is it the same with Bradely-Terry?

**Questions For Authors:**

The proposed preference embedding seems to map the last-layer hidden state through different heads to multiple scalars (similar to multi-head attention) and then sums them to obtain the final reward. It seems somewhat similar to ArmoRM[1]. However, each head in ArmoRM has a clear meaning, making the scoring more interpretable, while heads in GPM do not. Therefore, simply adding all scalars at the end of the forward pass makes me skeptical about its substantive significance.

[1] https://arxiv.org/abs/2406.12845

**Relation To Broader Scientific Literature:**

The primarily related domain is human preference alignment of LLM, where GPM can be useful in RLHF and supervised methods like DPO, SimPO, and so on.

**Theoretical Claims:**

I have checked the content in Sec 4 related to GPO formulation, which is correct and interesting.

---

> ### Author Rebuttal · Authors · 2025-04-01
>
> Thank you for your thoughtful and detailed review. We appreciate your constructive feedback and the opportunity to clarify aspects of our work. We address your points below:
>
> Below, we try our best to address the points you raised:
>
> 1. GPM Architecture, Training, & Efficiency:
> > Q1: The authors did not elaborate on the details of GPM's model architecture, and training objectives... hard to verify the claimed computational efficiency...
>
> - Architecture/Implementation: We describe the implementation of GPM in Section 4.2, detailing the eigenvalue scale gate and the eigenvector embedding head used to generate the preference embeddings.
> - Training Objective: The optimization objective for training GPM is explicitly defined in Appendix A.2 (Equation A.1). It involves minimizing the standard cross-entropy loss used in preference learning, based on the predicted preference probability ($\sigma(s(y_w>y_l∣x)$), where s is the GPM score from Eq 4.1) and the observed preference data $P_D(y_w>y_l∣x)$. This objective directly mirrors the typical objective used for training BT-based reward models.
> - Computational Efficiency: GPM achieves $\mathcal{O}(K)$ query complexity (Sec 4), matching BT models and improving on $\mathcal{O}(K^2)$ pair-based models, by processing responses individually.
>
> 2. GPM Performance on RewardBench:
>
> > Q2: Although GPM is claimed to have stronger expressiveness, it does not seem to show a consistent bonus on RewardBench...
>
> - Thank you for your observation regarding RewardBench performance. While performance can vary across specific sub-tasks and embedding dimensions (as shown in our ablation studies in Section 6.1 and Appendix E.1), GPM consistently outperforms the BT reward model on average across several tested base models, as shown in Table 2. Specifically, GPM showed average improvements of +7.44% (Gemma-2B-it), and +1.34% (Llama-3.1-8B-Instruct) over the BT baseline. Significant gains were often observed in the Chat and Chat-Hard categories. We appreciate the suggestion to analyze complex cases further and will add this to the appendix.
>
> 3. Clarity on Cyclic Preference Setting:
>
> > Q3: ...The remaining issue can be on cyclic preference, which I fail to understand the settings. A step-by-step explanation... would make it more clear.
>
> - These datasets were constructed using Ultrafeedback data, where we identified instances exhibiting cyclic preferences based on ratings across different criteria (e.g., helpfulness, honesty, instruction following). For example, a cycle might emerge where Response A is rated higher on honesty than B, B higher on helpfulness than C, but C higher on honesty than A. We provide details on the construction in Appendix E and results demonstrating GPM's near-perfect accuracy on these tasks in Section 6.2 and Table 1. Figure 3 also visualizes the learned embeddings for these cyclic cases.
>
> - We will add a more detailed step-by-step explanation of the dataset construction to the appendix in the revised version, as you suggested.
>
> 4. Preference Embedding Significance vs. ArmoRM:
>
> > Q4: ...similar to ArmoRM... each head in ArmoRM has a clear meaning... while heads in GPM do not. Therefore, simply adding all scalars...
>
> GPM's design prioritizes expressiveness and generality. The core idea is that complex, real-world human preferences might not always decompose neatly into a few predefined, interpretable scalar dimensions. They can exhibit intransitivity and context-dependency that require a more flexible representation.
>
> - Theoretical Grounding: Our multi-dimensional embeddings, combined with the skew-symmetric operator $\mathbf{R}^{\succ}$ and inner product $< \mathbf{R}^{\succ} v_{y_i|x},v_{y_j∣x}>$, provides a principled way to model any skew-symmetric preference relationship, including cycles. Theorem 1 and Theorem 4 provide theoretical grounding for this expressiveness.
> - Expressiveness and Generality: While ArmoRM's interpretable heads offer clarity, this structure inherently limits its capacity. Such models based on aggregating scalar rewards, even multi-dimensional ones, generally cannot capture arbitrary preference structures, particularly those involving intransitivity (like cycles) or complex mixtures of transitive and intransitive components.
> - Automatic Learning vs. Annotation: Furthermore, ArmoRM relies on aggregating scores from heads associated with predefined metrics (like helpfulness, truthfulness), which often require costly, explicit human annotation for each metric. In contrast, GPM aims to automatically discover and learn the relevant multi-dimensional preference representations directly from a single, standard pairwise preference signal ($y_w \succ y_l$). The dimensions in GPM are learned end-to-end to capture relevant preference factors without needing pre-defined semantics or multi-metric labels. The eigenvalue scale gate further allows the model to dynamically weight these learned factors based on the context.

---

### Official Review · Reviewer_awzY · 2025-03-10

**Overall Recommendation:** 3

**Summary:**

This paper introduces preference embedding, a novel approach to model human preferences for aligning foundation models that overcomes the limitations of traditional reward models like the Bradley-Terry model, especially in capturing intricate preferences. The authors propose the General Preference embedding Model (GPM), which embeds responses into a latent space, which is more expressive. Furthermore, the paper presents General Preference Optimization (GPO), a method that generalizes reward-based RLHF using the preference scores from GPM. Experimental results on RewardBench demonstrate that GPM consistently outperforms the BT reward model, particularly in modeling cyclic preferences. Evaluations on downstream tasks like AlpacaEval 2.0 indicate that aligning language models with GPO and GPM leads to performance improvements over methods using Bradley-Terry models.

**Claims And Evidence:**

Table 1 presents a comparison of Bradley-Terry (BT) reward models and General Preference embedding models (GPM) on cyclic preference
datasets. The results seem convincing and support the claim. The main problem is with Table 3, which shows the main result. It is very unclear. First of all, it's full of acronyms (e.g.  LC. WR, WR, Avg Len), and the acronyms are not explained in caption and main text. I suppose WR probably means "win rate", but it's also unclear to me it means the win rate over what. I can't understand the results unfortunately. This paper has obvious writing quality issues. This is the main reason to recommend rejecting this paper.

**Essential References Not Discussed:**

I'm not aware of critical misses. However, I think it'd be better to add references of more prior work in preference optimization, such as:

Mitchell, A note on DPO with noisy preferences & relationship to IPO, 2023
Liang et al., Robust preference optimization with provable noise tolerance for LLMs, 2024
Furuta et al., Geometric-Averaged Preference Optimization for Soft Preference Labels, 2024.

**Experimental Designs Or Analyses:**

Again, I don't fully understand the experiment settings because of writing issues.

**Methods And Evaluation Criteria:**

The method makes sense. However, as I explained previously, I don't fully understand the evaluation criteria because of writing issues.

**Other Comments Or Suggestions:**

I wasn't sure what (line 317) "We consider the iterative preference optimization process such as SPPO..." means. I suppose the authors actually means "GPO is a iterative preference optimization process, similar to SPPO..."?

**Other Strengths And Weaknesses:**

No additional comments

**Questions For Authors:**

Please revisit the presentation of the experiment section. I believe the whole thing needs to be rewritten.

**Relation To Broader Scientific Literature:**

It's related to RLHF and LLM preference optimization literature.

**Theoretical Claims:**

I don't find obvious issues in the theoretical claims.

---

> ### Author Rebuttal · Authors · 2025-04-01
>
> We sincerely thank the reviewer for their thoughtful and constructive comments. We especially appreciate the feedback regarding Table 3 and acknowledge that the acronyms used were not sufficiently explained in the current draft.
>
> 1. Clarification of Table 3:
> > Q1: First of all, it's full of acronyms (e.g. LC. WR, WR, Avg Len), and the acronyms are not explained in the caption and main text. I suppose WR probably means "win rate", but it's also unclear to me if it means the win rate over what. I can't understand the results, unfortunately.
>
> To clarify:
> - LC. WR stands for Length-Controlled Win Rate, a metric from AlpacaEval 2.0 that adjusts for length bias in generation.
> - WR denotes the Win Rate, i.e., the proportion of times our model's response was preferred over the baseline (GPT-4-turbo)'s, as judged by GPT-4-turbo.
> - Avg. Len refers to the Average Length of generated responses in tokens.
> The win rates are computed using pairwise comparisons between model outputs (e.g., our trained models vs. GPT-4-turbo) on the same set of prompts, with GPT-4-turbo as the evaluator.
>
> 2. Why These Acronyms Were Used:
> > Again, I don't fully understand the experiment settings because of writing issues.
>
> AlpacaEval is a widely used benchmark in nearly all recent LLM alignment papers (e.g., SimPO, SPPO, MagPie, Nemotron), and the format and metrics (including LC. WR) are standard in the literature. Due to space constraints, we opted to use the established abbreviations. That said, we agree that clearer exposition would benefit broader readability, and we will revise the caption and surrounding text to explicitly define these terms.
>
> 3. On the Role of This Table:
>
> > Q3: Table 1 presents a comparison of Bradley-Terry (BT) reward models and General Preference embedding models (GPM) on cyclic preference datasets. The results seem convincing and support the claim. The main problem is with Table 3, which shows the main result.
>
> We emphasize that our primary contribution is the development of a new, expressive architecture (GPM) for modeling general preferences, which addresses the limitations of traditional reward models like Bradley-Terry. The GPO-based alignment results in Table 3 are supplementary and serve to illustrate one possible downstream use case (Section 6.3). We believe that the clarity issues in Table 3, while important to correct, should not detract from the significance of our core contributions.
>
> 4. Additional Revisions:
>
> > Q4: I'm not aware of critical misses. However, I think it'd be better to add references to more prior work in preference optimization, such as:
> Mitchell, A note on DPO with noisy preferences & relationship to IPO, 2023
> Liang et al., Robust preference optimization with provable noise tolerance for LLMs, 2024
> Furuta et al., Geometric-Averaged Preference Optimization for Soft Preference Labels, 2024.
>
> We also appreciate the reviewer’s suggestions on related work. We will incorporate a discussion of the following relevant papers in the final version:
>
> Mitchell, A note on DPO with noisy preferences & relationship to IPO, 2023
>
> Liang et al., Robust preference optimization with provable noise tolerance for LLMs, 2024
>
> Furuta et al., Geometric-Averaged Preference Optimization for Soft Preference Labels, 2024.

---

### Official Review · Reviewer_gcRo · 2025-03-14

**Overall Recommendation:** 4

**Summary:**

This paper proposes *General Preference Embedding Model* (GPM) to improve LLM alignment on human preference.
The motivation is mainly on addressing limitations of the classical BT reward models such as challenges when facing intransitivity.
The authors deal with it by embedding the responses into latent space, introducing a skew-symmetric preference operator to derive preference scores.
This approach is novel in the context of LLM alignment and, by their empirical results on AlpacaEval2 and RewardBench, efficient and capable in capturing intransitive preference structures with linear computational complexity O(K).

**Claims And Evidence:**

The authors provide some theoretical results, which are personally appreciated, and empirical support.
In general I think the claims are solid.
But I will definitely appreciate further results on the generalisability of this novel method like for complex cyclic preference (please see 'weakness' below).

**Essential References Not Discussed:**

The references are adequate, yet some review on representation learning may be appreciated.

**Experimental Designs Or Analyses:**

It would be great if the authors could clarify whether additional robustness checks or sensitivity analyses were conducted, particularly regarding different user demographics or task variations.

**Methods And Evaluation Criteria:**

Experiments with cyclic preference datasets and RewardBench convincingly illustrate the performance advantages of GPM over BT models.
However, It would be beneficial to include additional context or examples from diverse, real-world scenarios (some benchmarks are not too hard to 'hack' by powerful models).

**Other Comments Or Suggestions:**

No more.

**Other Strengths And Weaknesses:**

Strengths:

1. As I stated before, I think the theoretical results and efforts should be appreciated, especially in the current LLM community.
2. the 'universal' framework which is inclusive of current methods (for example, k=1 -> BT model) is a good and inspiring idea. It can also deepen the understanding of the current popular methods.

Weakness:

1. While the method is indeed novel and I personally like the paper, I reserve my recommendation for a strong acceptance because tbh some results on these benchmarks may be challengeable, and more ablation and real-world scenarios are needed to make really strong claims, and some may argue that cute math often really doesn't buy you much.

**Questions For Authors:**

I'm really curious about the embeddings' dimension, like have you analyzed scenarios where embedding dimensions reflect competing or conflicting preferences? If such scenarios were encountered, how does the GPM manage these practical trade-offs?

**Relation To Broader Scientific Literature:**

This paper focused on RLHF but inspired by literature like preference learning and statistical methods.

**Theoretical Claims:**

I checked the theoretical claims and think they are clearly stated and easy to understand.

---

> ### Author Rebuttal · Authors · 2025-04-01
>
> We sincerely thank you for your time, insightful feedback, and constructive comments on our paper. We appreciate the recognition of GPM's novelty, theoretical grounding, and universal framework potential.
>
> We would like to address the specific points raised:
>
> >  Q1: Generalisability and further results on complex cyclic preferences.
>
> A1. We appreciate the suggestion regarding generalisability. As shown in Section 6.2/Table 1, GPM achieves near-perfect accuracy on cyclic preference datasets where BT models perform near-random guessing. This demonstrates GPM's effectiveness in capturing the intransitivity inherent in complex preference structures, a key limitation of BT models.
>
> > Q2: Need for more diverse, real-world scenarios/context, noting benchmark challengeability.
>
> A2. Thanks for your suggestion. We tested GPM on RewardBench (covering Chat, Chat-Hard, Safety, and Reasoning) and assessed downstream alignment using AlpacaEval 2.0. GPM consistently outperformed BT models across these benchmarks and different base models (Gemma-2B/9B, Llama-3.1-8B). GPM-integrated methods also showed improved win rates. While benchmarks have limitations, these consistent gains across diverse tasks suggest practical advantages.
>
> > Q3: More ablation/real-world scenarios are needed to make really strong claims.
>
> A3. We acknowledge your point that further studies could strengthen our claims. However, we believe the current results already provide significant support. The demonstrated superiority in modeling cyclic preferences, consistent outperformance on the diverse RewardBench tasks, and improvements in downstream alignment tasks collectively offer strong empirical evidence for GPM's effectiveness and advantages over traditional BT models. We have also included ablation studies on embedding dimensions (Section 6.1, Table 2) and GPM architecture design (Appendix E.1, Table 4).
>
> > Q4: Curiosity about embedding dimension analysis for competing/conflicting preferences and how GPM manages trade-offs.
>
> A4. Thanks for your feedback on analyzing embedding dimensions in scenarios with competing preferences and how GPM handles trade-offs. This is an excellent question that touches upon a core aspect of GPM's design. As discussed in Section 4.2, the multi-dimensional embedding space allows the model to automatically discover subspaces corresponding to various preference facets (e.g., helpfulness, honesty, style). The eigenvalue scale gate, which computes context-dependent eigenvalues {λ(x)}, then modulates the influence of these different dimensions based on the specific prompt. This mechanism allows GPM to dynamically weigh competing preference aspects and manage trade-offs based on context. Our ablation studies on embedding dimensions (Section 6.1, Table 2 and Appendix E.1, Table 4) further explore how performance varies with dimensionality across different tasks and models, indicating that the optimal configuration can depend on the specific trade-offs required.
>
> > Q5: Adding references on representation learning.
>
> A5. Thank you for suggesting a more extensive review of representation learning. We have acknowledged this connection briefly in Appendix D and agree that it's a relevant area. We will expand on this connection in the revised version.

---

### Decision · Program_Chairs · 2025-05-01

**Decision:**

Accept (poster)

**Comment:**

This paper proposes a generalization of Bradley-Terry Reward Models, in particular aiming at being able to handle intransitive human preferences, by embedding responses in a latent space. Empirical results on both synthetic data and common benchmarks show (1) higher accuracy of the resulting Reward Models and (2) higher quality of responses from models trained with them.

After the discussion period, all reviewers are leaning towards acceptance, praising in particular the novelty of the approach and the new light that it may shed on reward modeling (where BT is still king). However, several reviewers pointed out clarity issues, and questioned the practical relevance of the proposed method in real-world settings, where the prevalence of intransitive preferences is not obvious. Although authors provided clarifications as well as additional observations related to such intransitive preferences, I agree those are valid concerns that may limit the reach of this work.

In the end though, I am also leaning towards acceptance as I believe this novel take on preference modeling may inspire future developments to build more expressive reward models. I strongly encourage the authors to address reviewers' concerns regarding clarity in future revisions of their work.